

**Climatic subdivision of Heinrich Stadial 1 based on**
**centennial-scale paleoenvironmental changes observed**
**in the western Mediterranean area**
**Jon Camuera[1, 2], Gonzalo Jiménez-Moreno[2], María J. Ramos-Román[1], Antonio**
**García-Alix[2, 3], Francisco Jiménez-Espejo[3], Jaime L. Toney[4], R. Scott Anderson[5],**
**Cole Webster[5]**
[1] *Department of Geosciences and Geography, Faculty of Science, University of Helsinki, Finland*
[2] *Departamento de Estratigrafía y Paleontología, Universidad de Granada, Spain*
[3] *Instituto Andaluz de Ciencias de la Tierra (IACT), Consejo Superior de Investigaciones*
*Científicas-Universidad de Granada (CSIC-UGR), Granada, Spain*
[4] *School of Geographical and Earth Sciences, University of Glasgow, UK*
[5] *School of Earth and Sustainability, Northern Arizona University, USA*
Correspondence to: jon.camuera@helsinki.fi
Keywords: Climate, Heinrich Stadial 1, deglaciation, western Mediterranean, solar
activity
**ABSTRACT**
Heinrich Stadial 1 (HS1) is one of the most extreme climate periods of the last
glacial cycle, generating extremely low sea surface temperatures (SST) and significant
changes in terrestrial landscape (e.g., vegetation). Previous studies show that overall
cold/dry conditions occurred during HS1, but the lack of high-resolution records
precludes whether climate was stable or instead characterized by instability. A high-
resolution paleoclimatic record from Padul (southern Iberian Peninsula), supported by a
robust chronology, shows that climate during HS1 was non-stationary and centennial-



scale variability in moisture is superimposed on this overall cold climatic period. In this
study we improve the resolution and suggest a novel subdivision of HS1 in 7 sub-phases,
including: i) 3 sub-phases (a.1–a.3) during an arid early phase (HS1a; ~18.4–17.2 kyr
BP), ii) a humid middle phase (HS1b; ~17.2–16.7 kyr BP), and iii) 3 sub-phases (c.1–c.3)
during an arid late phase (HS1c; ~16.7–15.6 kyr BP). This climatic subdivision is
regionally supported by SST oscillations from the Mediterranean Sea, suggesting a strong
land-ocean relationship. A cyclostratigraphic analysis of our pollen data indicates that
HS1 climate variability, and thus this subdivision, is characterized by ~2000 and ~800-yr
periodicities, suggesting solar forcing controlling climate in this area.
**INTRODUCTION**

Understanding the background of natural climatic variability underlying abrupt

anthropogenic climate change is a main goal in paleoclimate research. In this respect,
deciphering rapid (e.g., millennial-scale) climate change and environmental impacts due
to Dansgaard/Oeschger (D/O) and Heinrich-like climatic oscillations during the last
glacial period and deglaciation have been the aim of ice, marine and terrestrial
paleoclimate investigations (Cacho et al., 2006; Höbig et al., 2012; Panagiotopoulos et
al., 2014).

Several paleoclimatic records evidenced the effect of especially cold and arid

conditions recorded during Heinrich Stadials (HSs) in marine and terrestrial environments
(Fletcher and Sánchez Goñi, 2008; Moreno et al., 2010; Martrat et al., 2014; Hodell et al.,
2017). High-resolution paleoclimatic records also show that climate during HS1 was
characterized by short-scale internal variability (Dupont et al., 2010; Stager et al., 2011;
Escobar et al., 2012; Zhang et al., 2014; Stríkis et al., 2015). However, few studies focus
on short-scale internal climate variability of HSs in southern Europe and the
Mediterranean region, and in particular within HS1 (Fletcher and Sánchez Goñi, 2008).



In this regard, a division of HS1 into two and three phases has previously been observed
in very few marine records (Supplementary Information and Table S1). Nevertheless, the
studies showing a three-phase division of HS1 disagree in the paleoenvironmental
characterization of each phase (Table S1) and a complete knowledge of the variability
within HS1 has yet to be achieved (Hodell et al., 2017).

Here, we present pollen and sedimentation data between 20 and 11 kyr BP from

the new Padul-15-05 terrestrial sedimentary record (southern Iberian Peninsula; Fig. 1),
registering regional and local paleoenvironmental responses to climate changes during
HS1 and deglaciation, i.e., Bølling-Allerød (BA) and Younger Dryas (YD). The high-
resolution data (~61-yr) from 18.4 to 15.6 kyr BP revealed centennial-scale variability
during HS1, which is replicated in other Mediterranean paleoclimatic records and enables
us to suggest, for the first time, an accurate internal climatic subdivision of HS1.
**MATERIALS AND METHODS**

In this study we used 10 AMS radiocarbon dates to obtain an accurate

chronological control between 20 and 11 kyr BP from the Padul-15-05 record (Fig. 2a
and Table S2; Supplementary Information for more precise methodology).

The Mediterranean forest, xerophytes, Pollen Climate Index (PCI) and

Precipitation Index ($I_p$) were used as pollen paleoclimatic proxies (Fig. 2c-f and Fig. 3a).
The PCI is useful for temperature and precipitation related climate changes, whereas $I_p$ is
a proxy for precipitation reconstruction in this region. In addition, normalized silicon
($Si_{norm}$) data from XRF analysis (Fig. 2b) was used as indicator of the siliciclastic input
from the Sierra Nevada into the wetland (Camuera et al., 2018) (Methods, Supplementary
Information).

For the purpose of identifying cyclicities related to regional climate oscillations,

a cyclostratigraphic spectral analysis using the REDFIT procedure under rectangular



window function from (Schulz and Mudelsee, 2002) was performed on xerophyte and $Ip$
data, which have been proven to be good proxies for regional moisture availability in this
area (Pini et al., 2009; Fletcher et al., 2010). A spectral analysis was also run on raw GRIP
$^{10}$Be flux data between 18.6 and 11 kyr BP (Adolphi et al., 2014) in order to observe
cyclicities related to solar activity (Fig. 4; Methods, Supplementary Information).
**RESULTS AND DISCUSSION**
**Heinrich Stadial 1 (HS1)**

The terrestrial paleoclimate record from Padul shows overall cold and arid

conditions during HS1, deduced by the decrease in mesic forest and abundance of
xerophytes between 18.4 and 15.6 kyr BP (Fig. 2c, d). Centennial-scale variability is also
observed during HS1 that can be divided into 3 main climatic phases (i.e., HS1a from
18.4 to 17.2 kyr BP, HS1b from 17.2 to 16.7 kyr BP, and HS1c from 16.7 to 15.6 kyr BP)
and a further subdivision in 7 smaller-scale phases within them (i.e., HS1a.1, HS1a.2,
HS1a.3, HS1b, HS1c.1, HS1c.2 and HS1c.3) (Fig. 2e, f and Fig. 3a).

The first of the three main climatic phases in Padul, HS1a (early HS1; 18.4–17.2

kyr BP), is characterized by low temperatures with significant variability in precipitation
but under generally arid conditions, deduced by high xerophytes and low PCI and $Ip$
values (Fig. 2c, e, f). Especially cold/arid conditions during this early phase are confirmed
by high $Si_{norm}$ values, which show that high siliciclastic input from the Sierra Nevada
range into the wetland are caused by enhanced erosion during decreased forest cover
(Camuera et al., 2019). The general cold/arid conditions shown in Padul during the early
HS1a were also documented in nearby marine records presenting the 3 main phases for
HS1, such as the pollen records from NW Iberia (Naughton et al., 2016), or the pollen
data, SST reconstructions and foraminifera/coccolithophore assemblages from Alboran
Sea (Fletcher and Sánchez Goñi, 2008; Martrat et al., 2014; Bazzicalupo et al., 2018).



HS1b (middle HS1; 17.2–16.7 kyr BP) is characterized in Padul by a moderate
increase in temperature and precipitation, deduced by low xerophytes, and higher PCI
and *Ip* values. This is further supported by low $Si_{norm}$, indicative of low erosion precluding
siliciclastic input in the wetland (Fig. 2b, c, e, f). A similar slightly warmer climate during
this phase was recorded in SST records from the Mediterranean (Cacho et al., 1999;
2006), and in particular, from Alboran Sea (Martrat et al., 2014) (Fig. 3b and c, subjected
to age uncertainties for onset/ending of HS1). This warmer/wetter conditions agree with
increases in temperate forest recorded in the Iberian margin (Daniau et al., 2007), and in
runoff in Lake Estanya (NE Spain) (Morellón et al., 2009).
HS1c (late HS1; 16.7–15.6 kyr BP) was climatically similar to HS1a,
characterized by cold/dry conditions. This is deduced by the observed increase in
xerophytes and $Si_{norm}$ and lowering in *Ip* between ~16.9 and 15.8 kyr BP, related with the
decreasing moisture (Fig. 2b, c, e). The general cold/arid climate in Padul during this
phase is concordant with low SST from Alboran Sea (Martrat et al., 2014) (Fig. 3a, b),
and with increasing salinity and low lake level in Lake Estanya (Morellón et al., 2009).
Our high-resolution record also revealed shorter centennial-scale climatic
variability during HS1a and HS1c with a further climatic subdivision of HS1 into 7 sub-
phases (i.e., HS1a.1, HS1a.2, HS1a.3, HS1b, HS1c.1, HS1c.2 and HS1c.3):
HS1a.1 was characterized by a cold/arid phase between 18.4 and 17.8 kyr BP
recorded by high xerophytes, and low PCI and *Ip* values. Climate changed towards more
humidity in HS1a.2 sub-phase at 17.8–17.5 kyr BP, and returned to enhanced aridity
during HS1a.3 between 17.5–17.2 kyr BP. This arid-humid-arid climatic pattern is further
confirmed by oscillations in $Si_{norm}$ (Fig. 2b, c, e, f and Fig. 3a).
HS1c also presents a three-phase subdivision, namely HS1c.1, HS1c.2 and
HS1c.3. HS1c.1 was characterized by a decrease in precipitation and temperature (low *Ip*



and lowest PCI values), registering the coldest conditions of HS1 at 16.7–16.4 kyr BP
(Fig. 2f). Temperature and moisture conditions increased during HS1c.2 at 16.4–16 kyr
BP, whereas similar temperatures but under more arid climate conditions are recorded
during HS1c.3 at 16–15.6 kyr BP (Fig. 2c, e, f and Fig. 3a). This arid-humid-arid climatic
pattern is similar to the earlier HS1a.

Environmental changes recorded in Padul represent centennial-scale climate

oscillations during HS1, which can be correlated with other regional records. The
centennial-scale arid-humid-arid trends recorded during HS1a and HS1c, and the increase
in temperature/precipitation during HS1b, are also observed in the SST records from the
Alboran Sea in western Mediterranean (Cacho et al., 1999; 2006; Martrat et al., 2014)
and in the GISP2 ice core (Grootes et al., 1993) (Fig. 3a-d), suggesting a similar response
in continental, marine and ice sheet environments to climatic forcing (see section below).
The presented age offsets between records could be related with variations in reservoir
ages of the Atlantic and Mediterranean promoted by thermohaline circulation collapse in
both areas during HS1 (Sierro et al., 2005). In addition, the significant decrease in the
atmospheric [14]C between 17.5 and 14.5 kyr also difficult age models during this period
(Broecker and Barker, 2007), whereas dating on different foraminifera species can also
produce large differences on radiocarbon ages, especially during HS1 (up to 1000 years)
(Ausín et al., 2019). The asynchronicity and the early record of HS1 in Padul (18.4–15.6
kyr BP) with respect to the equivalent GS-2.1a cold event from Greenland ice-core
records (17.5–14.7 kyr BP) is evident. This could be due to different environmental
responses consequence of the different latitude and geographical features between high-
latitude Greenland and mid-latitude Mediterranean and Iberian records. An early record
of HS1 in the study area is supported by the well-dated paleoenvironmental records from
our region. For instance, the growth interruption of the CAN speleothem (N Spain)



occurred between 18.2 and 15.4 kyr BP (Moreno et al., 2010) supporting an early HS1
(or Mystery Interval in their study), the increasing moisture conditions in Lake Prespa
(Macedonia, Albania, Greece) at 15.7 kyr BP indicating an early end of HS1 (Cvetkoska
et al., 2015), and the pollen record from MD99-2331 (NW Iberian margin), which
evidence HS1 between 18.8 and 15.8 kyr BP (Sánchez Goñi et al., 2018).
Despite the offsets of SST reconstructions from Mediterranean Sea,
environmental oscillations in both areas should have been synchronous. Therefore,
warming peaks recorded in Padul and in SST records during HS1 were coetaneous, result
of the strong land-ocean interaction (Sánchez Goñi et al., 2018).

**Bølling-Allerød (BA) and Younger Dryas (YD)**

The BA recorded in Padul between 15.6 and 12.9 kyr BP is characterized by
significant increase in the Mediterranean forest, and thus *Ip* and PCI values, indicating
warmer/wetter climate than during HS1. In addition, Padul is one of the few continental
records to detect the 5 centennial-scale sub-phases during the BA, similar to the GI-1e to
GI-1a from Greenland ice cores (Johnsen et al., 1992; Grootes et al., 1993) (Fig. 2e, f and
Fig. 3d).
Cold/arid climate during the YD stadial also affected the paleoenvironments in
this area between 12.9–11.6 kyr BP. The YD is characterized by relatively arid conditions
show by the mean xerophyte percentage of ~22%, whereas temperature seems to increase
throughout the YD period, reflected by the increasing trend in the Mediterranean forest
(Fig. 2c, d).

**Climate variability and solar forcing in the Iberian Peninsula**

Centennial- and millennial-scale climate variability have been recorded during
HS1, BA and YD period in Padul. The spectral analysis on xerophytes and *Ip* presented
~2000, 800, 500 and 200-yr cycles (Fig. 4a-c). These climatic variabilities could be





related to solar forcing, as similar cyclicities have been obtained analyzing solar activity
with [14]C production rates (Damon and Jirikowic, 1992; Turney et al., 2005). Several
studies have determined a relation between paleoenvironmental data oscillations linked
to climate changes through variations in solar activity (Bond et al., 2001; Lüning and
Vahrenholt, 2016). In particular, climate variability during the Last Glacial and Holocene
periods was strongly controlled by solar activity, specifically during cold glacial phases,
in which solar variability caused larger climate changes (van Geel et al., 1999). In
addition, more recent temperature estimations showed that they also seem to be forced by
solar variability (Soon et al., 2015).

The   obtained   ~2000-yr   climatic   cyclicity   forced   millennial-scale

paleoenvironmental variability in Padul and permitted the three-phase division of HS1
(Fig. 5a, b). This cyclicity could be linked to D/O-like variability, which presents a 1–2
kyr periodicity during the last glaciation (Bond et al., 1999), such as the ~1.8-kyr cycle
identified on the hematite-stained grain record from North Atlantic cores (Bond et al.,
1999). Paleoclimatic records from the Equator and Southern Hemisphere also determined
periodic surface temperature variations of around 2000 yrs in relation with solar
irradiance (Bütikofer, 2007).

The ~800-yr cycle identified in Padul forced the centennial-scale climatic

subdivision of HS1 in 7 sub-phases (Fig. 5a, c). A similar ~800 yr cyclicity characterizes
the [10]Be flux data indicative of changes in solar activity (Adolphi et al., 2014) (Fig. 4d
and Fig. 5d, e). Other global paleoclimatic studies also show similar frequencies caused
by solar variability. For example, an ~800-yr cycle was observed in Irish oak tree
chronologies (Turney et al., 2005) and in Mg/Ca SST from the Pacific Ocean (Marchitto
et al., 2010), both records closely related to solar irradiance. A 890-yr cycle was also
found in the $\delta^{18}O$ Holocene time series from Greenland and interpreted as linked to solar



radiation (Schulz and Paul, 2002). Consequently, the ~800-yr cycle detected in Padul and
in other worldwide records suggests a linkage between centennial-scale
paleoenvironmental changes and solar activity.

The data from Padul display a good correlation between environmental changes

in the southern Iberian Peninsula and Mediterranean SSTs during HS1 (Fig. 3a-c),
suggesting a close land-ocean relationship in response to solar variability. The
Mediterranean SST could have been affected by solar activity, similar to the North
Atlantic cooling episodes linked to reduced solar irradiance (Bond et al., 2001). In
addition, observed variations in the Padul data suggest a southward shift of the North
Atlantic polar front during HS1 (Repschläger et al., 2015), which could have produced a
penetration of colder Atlantic surface waters into the Mediterranean (Cacho et al., 1999;
Sierro et al., 2005). These conditions, along with the southward displacement of the North
Atlantic atmospheric polar front, could have produced a low land-ocean temperature
contrast and weak moisture advection between both environments, and therefore,
increasing aridity in the western Mediterranean during cold sub-phases HS1a.1, HS1a.3,
HS1c.1 and HS1c.3. Similar conditions linked to weak moisture advection were
interpreted in the eastern Mediterranean during HS1 and HS2 (Kwiecien et al., 2009) and
in the Corchia Cave during HS11 (Drysdale et al., 2009). In contrast, during warmer sub-
phases in Padul (i.e., HS1a.2, HS1b and HS1c.2) and in Mediterranean SSTs, enhanced
marine evaporation and moisture advection toward the continent could have provoked
wetter climate conditions in southern Iberian Peninsula.
**CONCLUSIONS**

The high-resolution analysis of the Padul-15-05 continental record for the 20–11

kyr BP interval shows that:





1) Centennial-scale climate oscillations affected southern Iberian Peninsula
during HS1, with three main phases HS1a (18.4–17.2 kyr BP), HS1b (17.2–
16.7 kyr BP) and HS1c (16.7–15.6 kyr BP) characterized by general
arid(cold), humid(cool) and arid(cold) climate, respectively.
2) We suggest for the first time a further subdivision within these 3 main climatic
phases of HS1 in 7 sub-phases: 3 sub-phases (a.1–a.3) during HS1a, HS1b,
and 3 sub-phases (c.1–c.3) during HS1c. The climatic variability is also
identified in Mediterranean SST records, confirming this climatic pattern at
regional-scale.
3) The main periodicities obtained for climatic oscillations of ~2000 and ~800
yrs within HS1, BA and YD seem to be related to solar forcing. Variations in
solar activity could have influenced latitudinal shifts of the North Atlantic and
atmospheric polar fronts, affecting the land-ocean temperature contrast,
marine evaporation and moisture advection toward the continent.
**ACKNOWLEDGMENTS**
This research is supported by the projects CGL2013-47038-R and CGL-2017-
85415-R, PhD funding BES-2014-069117 (Jon Camuera) and Ramón y Cajal fellowship
RYC-2015-18966 (Antonio García-Alix) provided by the Ministerio de Economía y
Competitividad of the Spanish Government. Additional funding was provided by the
research group RNM0190 and the project P11-RNM-7332 with a postdoctoral fellowship
(María J. Rámos-Román) from the Junta de Andalucía.
**DATA AVAILABILITY**
The paleoclimatic pollen data from Padul-15-05 can be found in the PANGAEA
data repository (https://doi.pangaea.de/10.1594/PANGAEA.904053, dataset *in review*).



## AUTHOR CONTRIBUTIONS

J.C. performed the pollen, XRF and spectral analyses, interpreted the data and wrote the manuscript. G.J.-M. discussed data and interpretations and wrote the manuscript. M.J.R.-R. performed the XRF analysis, discussed data and interpretations and contributed to the writing of the manuscript. A.G.-A., J.L.T., R.S.A., F.J.E. and C.W. discussed data and interpretations and contributed to the writing of the manuscript.

## COMPETING INTEREST

The authors declare no financial and non-financial competing interests.

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



**FIGURES AND FIGURE CAPTIONS**

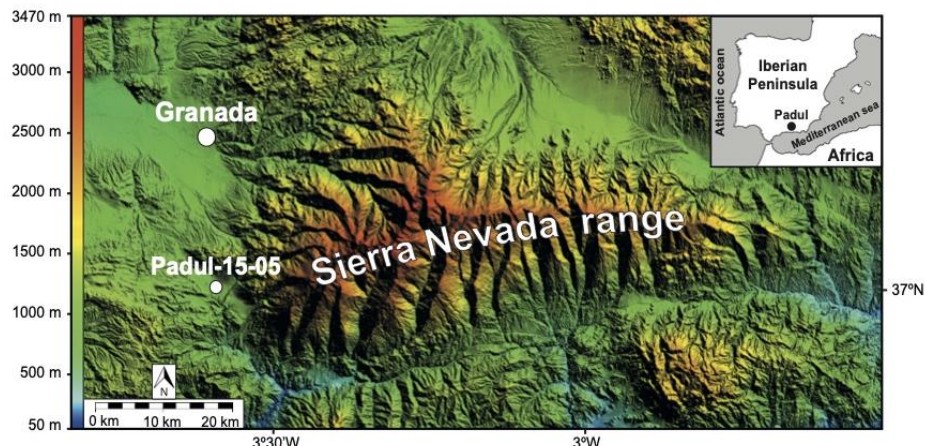


**Figure 1.** Geographical location of the Padul-15-05 record in the western margin of Sierra
Nevada range and south of Granada city (southern Iberian Peninsula) (modified from
Camuera et al., 2018).

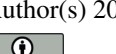



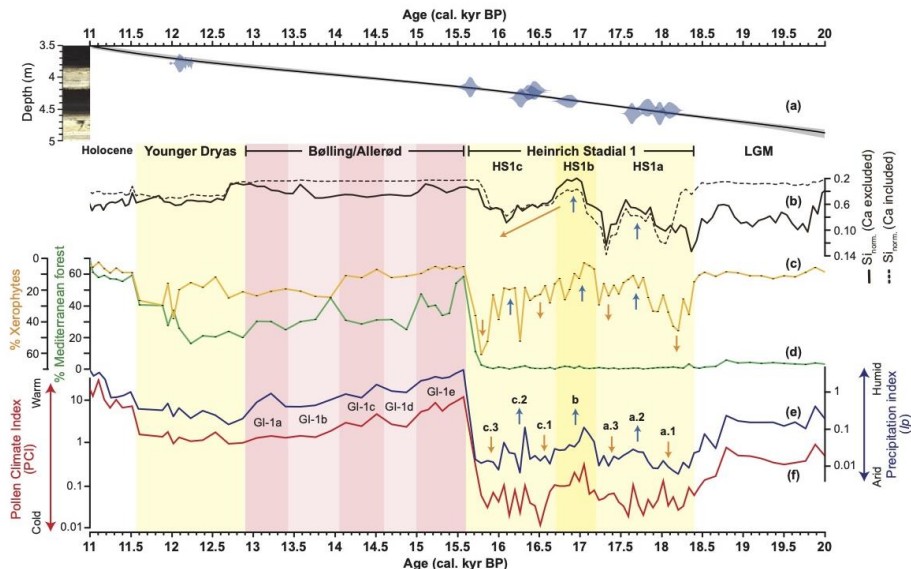

**Figure 2.** Paleoclimatic raw data from the Padul-15-05 sediment core for the time period between 20 and 11 kyr BP: (a) Photograph and age-depth model. (b) Normalized silicon values, with calcium excluded (continuous line) and included (dashed line) from total counts (values inverted) (Methods, Supplementary Information). (c) Percentage of xerophytes (values inverted). (d) Percentage of Mediterranean forest. (e) Precipitation Index (*Ip*). (f) Pollen Climate Index (PCI). Yellow shadings show the Younger Dryas (YD) and Heirich Stadial 1 (HS1). Dark yellow shading within HS1 indicates the slightly warmer/wetter middle phase (HS1b). Red and pink shadings show the Bølling-Allerød (BA). In particular, red shadings correspond to the warmer/wetter Greenland Interstadials 1a, 1c and 1e, and pink shadings to the colder/more arid Greenland Interstadial 1b and 1d. Blue arrows indicate moderately warmer/wetter sub-phases within HS1, whereas orange arrows show colder/more arid sub-phases.



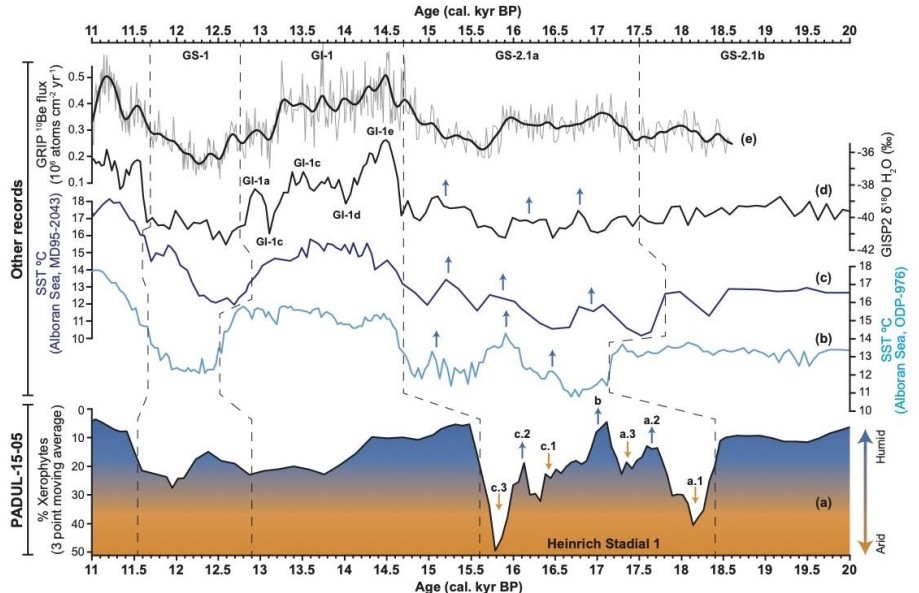

**Figure 3.** Paleoclimatic proxy data from Padul, Mediterranean Sea and Greenland ice-cores for the time period between 20 and 11 kyr BP: (a) Xerophyte data from Padul-15-05 with three-point moving average (values inverted). (b) SST (degrees Celsius) from ODP-976 record of Alboran Sea (Martrat et al., 2014). (c) SST (degrees Celsius) from MD95-2043 record of Alboran Sea (Cacho et al., 1999; 2006). (d) Raw $\delta^{18}$O H$_2$O values (‰) from GISP2 (Grootes et al., 1993). (e) Raw $^{10}$Be flux values ($10^6$ atoms cm$^{-2}$ yr$^{-1}$) (grey line) and smoothed data (black line) from GRIP (Adolphi et al., 2014). Within HS1, blue and orange arrows in the Padul record show the humid and arid phases, respectively. In the SSTs from Alboran Sea and the Greenland record during HS1, blue arrows marked the warmer temperatures in relation with the relatively more humid phases from Padul for this period (HS1a.2, HS1b and HS1c.2). Vertical dashed lines show transitions between LGM-HS1 (GS-2.1b – GS-2.1a), HS1-BA (GS-2.1a – GI-1), BA-YD (GI-1 – GS-1) and YD-Holocene for each study.





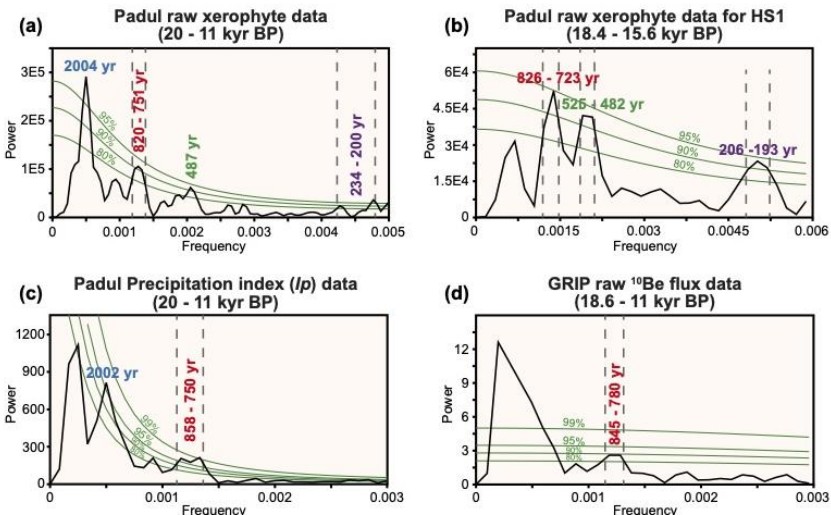

**Figure 4.** Cyclostratigraphic analysis of the Padul-15-05 pollen data and Greenland ice-core record during HS1. Spectral analysis run on: (a) Raw xerophyte percentages from the Padul-15-05 record for the age range between 20 and 11 kyr BP. (b) Raw xerophyte percentages for HS1 (18.4 – 15.6 kyr BP). (c) *Ip* data for the age period between 20 and 11 kyr BP. (d) GRIP [10]Be flux data between 18.6 – 11 kyr BP. Note that the spectral peak of the GRIP [10]Be between 0.0001973 and 0.0005919 frequencies (a cycle with a periodicity between 5068 and 1689 years) seems to be an artefact, as the longer periodicity cycles (closer to 5 kyr) cannot be significant in a time series of data spanning 7600 years.

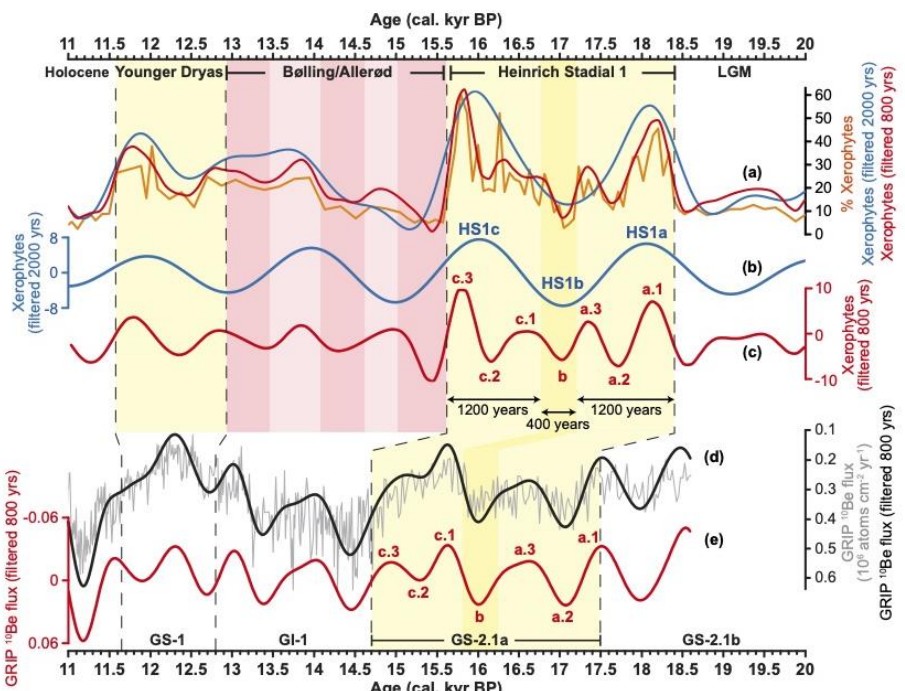

475

**Figure 5.** (a) Raw percentages of xerophyte taxa (orange line) along with the filtered

xerophyte taxa based on the obtained ~2000-yr cycle (blue line) and the ~800-yr cycle

(red line) (see spectral analysis from Figure 4a and b). (b) Xerophyte data filtered using

a bandwidth parameter of 0.0001 for the ~2000-yr cycle (blue line). (c) Xerophyte data

filtered using a bandwidth parameter of 0.0006 for the ~800-yr cycle (red line). Note that

the 3 main phases (HS1a, HS1b and HS1c) are marked within HS1 in relation with the

~2000-yr cycle, and the internal sub-phases (a.1-a.3, b, and c.1-c.3) in relation with the

~800-yr cycle. The length of the HS1a, HS1b and HS1c have also been marked. (d) GRIP

[10]Be flux data (10[6] atoms cm[-2] yr[-1]; values inverted) (grey line) filtered to the obtained

cyclicity of ~800 yr (black line) (see spectral analysis from Figure 4d). (G) GRIP [10]Be

flux data filtered using a bandwidth parameter of 0.0006 for the ~800-yr cycle. Vertical

dashed lines have been marked according to the Greenland Stadials (GS-2.1b, GS-2.1a,

GS-1) and Interstadial (GI-1) delimitations (Rasmussen et al., 2014).