# Peer review of "in the western Mediterranean area"

_Climate of the Past, 2019_

## Short Comment (SC1) · 15 Dec 2019

This is an impressive set of data; I have seen it at conferences and I was looking forward to seeing it published, but this work leaves my professional curiousity unsatisfied and disappointed. The authors aim to shed light on the centennial climate variability in the Western Mediterranean region during the late glacial and document a three-phased nature of HS1. In fact, they fail. The new high-resolution data are exciting and have great potential, but the interpretation is somewhat careless. E.g.: I do not see how the HS1 in Spain could lead the hemispheric (?) signal by ca. 1 ka? This is a bald and

provocative statement and the authors suggest "different environmental responses" (lines 144) to remedy this problem. This argument is not convincing, in particular, without explaining and elaborating on the nature of the purported responses. Perhaps the answer lies in the uncritical approach to the Padul sequence 14C-based chronology? Independent of how good the age model is – it is just a model. A critical discussion of potential flaws of the Padul chronology and their implications might solve the problem. Also, while the authors document the three-phased nature of HS1 they discuss neither the causes nor regional implications of the described features. Why does the division of HS1 matter at all? Further, The authors call for solar activity as a driver of changes in the Padul proxies. Can they elaborate on the exact mechanism? How solar activity translates to floral assemblage changes? Last but not least, the spectral analyses results seem a bit at odds with common sense. 800 yr cyclicity during HS1 event of less than 3 ka duration is already suspicious. Periodicity of 2000 yr within HS1, BA and YD (line 232-233) is simply absurd! The YD itself only lasted for ca. 1000 yr. Paleoclimate research is so much more than tuning wiggly curves and finding prescribed periodicities in proxy records! Valid, original observations call for careful and thorough and original interpretations rather than preaching to the choir using empty but catchy phrases and unsubstantiated claims.

I wish the authors will address mentioned points in the revised version of their manuscript.

best regards Ola Kwiecien

---

## Referee Comment (RC1) · Anonymous Referee #1 · 24 Jan 2020

General comments

The manuscript by Jon Camuera and colleagues describes paleoenvironmental changes in southern Iberia during the last deglaciation focusing on Heinrich Stadial 1 (HS1). The authors observe a novel subdivision of HS1 in the analyzed Padul record and in other records from the Western Mediterranean and Greenland. They come to the conclusion that solar forcing accounted for an detected ∼2000 and ∼800 yrs climate cyclicity.

[Figure]

The study presents novel ideas and addresses relevant questions within the scope of the journal Climate of the Past. It is well structured, easy to follow, and concisely written. Figures are of good quality.

However, I have two main concerns. Firstly, it is not always clear whether data is new, already published, or already published but analyzed/shown in a new way. This concerns mainly the own previous studies. Nevertheless, it is important to exactly indicate the sources to avoid (self-) plagiarism (see also specific comments).

Secondly, the age-depth model is not as robust as stated. That does probably not affect the observed climate pattern but it may affect the cyclostratigraphic analysis. The age-depth model is based on radiocarbon dates of organic bulk sediments that might need a reservoir correction. Particularly in a wetland setting, a reservoir age of aquatic and semi-aquatic plants must be considered. The uncertainties of the age-depth model need to be taken into account and should be critically discussed when correlating records and when analyzing cyclicities.

Specific comments

23/62: Please relativize the terms "robust" and "accurate".

25: Please clarify which resolution is improved.

34/35: Why does natural climatic variability underlie abrupt anthropogenic climate change? Please clarify or rephrase.

55: Please mention the section "Regional and Local Settings" of the Supplementary Information here. In addition, please delete "new" to avoid misunderstanding.

62–64: Please add reference of the radiocarbon dates.

65/66: Please add reference of the pollen data, e.g. add "based on palynological data by..." after "Precipitation Index (Ip)". If I understood it right, the palynological data has already been published, but it is not clearly stated in the manuscript.

[Figure]

81–83: How is the start (lower boundary) of HS1 defined in your record? Could it have also started at ca. 18.7 kyr when Si, Mediterranean forest, PCI and Ip start to decline?

97/98: Please add "(Fig. 3b, c)" after SST reconstructions and "Cacho et al., 1999; 2006" to the references.

102–104: Please rephrase the sentence because the SST records published by Cacho et al., 1999; 2006 originate from the Alboran Sea as well.

109–111: Please add PCI because it shows the same pattern.

112: Please replace "(Fig. 3a, b)" by "(Fig. 3b)".

136–157: The presented explanations and records are not strong enough to conclude an early HS1 and Bølling-Allerød in the Mediterranean.

159–169: Please add comparisons with other regional records.

166–169: Xerophytes decrease at first. How can that be explained? How is the lower boundary of the YD defined in your record? Better mention the Ip value to suggest arid conditions. In general, it would be nice to see a detailed pollen diagram in the supplement to comprehend the stated climate variations.

185–188: The D/O-record for 20-11 kyr is well defined and does not show a ∼2 kyr cyclicity.

244: I appreciate that you provide the data in an online repository. However, I suggest adding the complete palynological dataset to the repository for replicability.

Figure 2–5: Please indicate all sources of data.

Figure 2a: The uncertainty of the age-depth model is underestimated where no dates are available. Please use a model that accounts better for uncertainties. In addition, please add the dates that you rejected to Fig. 2a, e.g. in a different color.

Figure 4: I suggest to use always "xerophyte percentages" instead of "raw xerophyte

data" and "raw xerophyte percentages" (also in Supplementary Information line 120). In addition, please indicate the meaning of the green lines (confidence interval) in the figure caption. Which periodicity is shown by the first peak in Fig. 4d and why is it not mentioned?

Supplementary Information (SI): The Supplementary Information is a rather extensive compilation of additional details. I appreciate the methodological details here. However, I suggest including the previous studies to the main text because they contain important data for comparison. For an even better comparison, I suggest adding this study to table S1.

Table S2: Please add source (reference or this study) to each date.

SI 54: Please add one or two sentences about the recent vegetation.

SI 91–93/100: Please indicate which taxa are mesothermic and which are steppic.

SI 107–109: Is this new or already published data? Please clearly indicate.

SI 120–125: Which parameters were used for the Ip analysis? Could you add Ip to the first sentence?

SI 120–137: Why were exactly these datasets used? Why is there only one analysis for HS1?

Technical corrections:

74: Please edit format of reference.

167: shown.

SI 125: Please add "(CI)" after "Confidence Interval".

SI 129: analyses.

---

## Referee Comment (RC2) · Anonymous Referee #2 · 3 Feb 2020

This paper presents the details of the recently published new Padul pollen record for the Heinrich Stadial 1 and Lateglacial interval.

The pollen record reveals significant changes over the study interval, presented in the form of pollen-based indices with established use in the study area. The record is also at a high temporal resolution offering new insights into centennial-scale variability during the study interval.

There are fascinating visual parallels between the pollen indices for the Heinrich Stadial

1 interval and SST records from the nearby W Mediterranean (Alboran Sea) which generally support the interest and interpretation of rapid climate variability during this interval.

The main difficulty for the manuscript is the chronology of the record. In essence, the Heinrich Stadial appears "too old" in the Padul record, and this creates difficulties for the analysis and interpretation. The manuscript seems to have a "split personality" – attempting to interpret both (A) the difference in ages between the Padul record and other records as a real and meaningful phenomenon, e.g. with implications for reservoir ages, etc. and (B) propose synchroneity of events between Greenland and S Iberia, e.g. as shown by wiggle-matched records in some figures and direct labelling of pollen changes with Greenland event stratigraphical terminology. It should be noted that conceptually (A) and (B) are mutually exclusive and they sit together very uncomfortably in the manuscript.

Regarding (A), the authors suggest that changes in marine reservoir effects might explain the difference in apparent age of the Heinrich stadial between Padul and the Iberian margin records. However, the logic is reversed here and the apparently older age of the Padul record cannot be explained away by marine reservoir effects which would tend to give older ages in the marine realm, not the terrestrial. Furthermore, the study of coupled land-sea tracers in nearby Alboran records (Comboureiu Nebout et al., 2009; Fletcher et al., 2010) already reveals a synchronous (within age model uncertainty) coupling of climate changes over the W Mediterranean and the high-latitudes, with possible modest enhancements of up to ∼200 years of the marine reservoir effect.

Overall, I suspect that there are uncertainties in the site-specific age model which are not dealt with fully in the manuscript. Essential information for the validity of this study about stratigraphy, age control data and rejected dates must be included and discussed in the main manuscript and not placed in the supplementary material. Inspecting the radiocarbon data, it is evident that there are difficulties with reservoir ages or old carbon sources leading the authors to reject several dates obtained on bulk carbonates and

gastropods. However, I do not see that it can be excluded that old carbon effects are not impacting also on the included dates made on bulk sediment. The authors need to deal with this more directly in the presentation of the record and ultimately the interpretation of the data. If the uncertainties in the age model are too great to support (A) then this shortcoming should be accepted and the implications of (B) can still be tentatively explored.

Without a more open and direct appraisal of the age control issue, I do not think that the time series analysis can be sustained. Although there do appear to be interesting pseudo-cyclical patterns in the proxies for some time intervals, the authors must be cautious about over-interpreting weak spectral signals (e.g. at 80%, 90% confidence levels) and cautious about identifying spectral peaks at high frequencies occurring close to three times the sampling resolution which may be spurious.
* * *

---

## Author Comment (AC1) · 3 Feb 2020

Response to comments by Ola Kwiecien: "Comment on the manuscript"

This is an impressive set of data; I have seen it at conferences and I was looking forward to seeing it published, but this work leaves my professional curiosity unsatisfied and disappointed. The authors aim to shed light on the centennial climate variability in the Western Mediterranean region during the late glacial and document a three-phased nature of HS1. In fact, they fail. The new high-resolution data are exciting and have great potential, but the interpretation is somewhat careless. E.g.: I do not see how the HS1 in Spain could lead the hemispheric (?) signal by ca. 1 ka? This is a bald and provocative statement and the authors suggest "different environmental responses" (lines 144) to remedy this problem. This argument is not convincing, in particular, with-out explaining and elaborating on the nature of the purported responses. Perhaps the answer lies in the uncritical approach to the Padul sequence 14C-based chronology? Independent of how good the age model is – it is just a model. A critical discussion of potential flaws of the Padul chronology and their implications might solve the problem.

We thank the reviewer for her excellent comments, and we have endeavored to incorporate them into our thoughts and into the manuscript, as well as provide explanations where we either were unclear, or where we disagree.  It is important to remember that, while one outcome of this analysis is that our age model suggests an early beginning to HS1 from the Padul data, the primary goal of this analysis was *to investigate the characteristics of potential subdivisions of the HS1 as expressed in the Padul record*. With this in mind, we address the comments of the reviewer directly below:

We agree that there are limitations in radiocarbon dating. However, in the Padul-15-05 sedimentary record we tried to reduce age uncertainties for the last 20,000 years by providing a total of 27 radiocarbon samples (including 2 radiocarbon samples on specific compounds) (Camuera et al., 2018).  More specifically for the HS1 interval, we have 10 AMS radiocarbon dates, ranging over an ~2400 year interval, between 17,980 and 15,658 cal yr BP (see Figure 2a and Table S2). Even though we believe that this is a very good spread of ages for this time period, one potential flaw in our age model is that two short gaps of less than 500 years appear in the probability distributions of this sequence (Table S2).  However, combined with the fact that 9 of the 10 ages in this interval are in stratigraphic order suggests that this age model has considerable power for the entire interval.

Further, in order to strengthen our model and prevent circularity, we have not tuned our model to any other paleoclimate records, which would have eliminated the capacity to describe succession of events and pre-conditioning triggers. We were surprised by the results we obtained – that of an earlier record of HS1 from the Padul sediments – when compared to the other paleoclimate records.  Our results then became the impetus for the current manuscript.

Nowhere in the paper did we mention that the HS1 in Spain led the hemispheric signal. We apologize if our explanation was not clear enough. Here we wanted to show that in southern Iberian Peninsula (where the Padul record is located) environmental signals associated with the HS1 (e.g., high aridity) preceded the age established for this event in other records located at higher latitudes. This is an outcome of our age model,

which we believe, but it is not the primary object of this analysis. Even so, several previous studies support an earlier record of deglaciation and asynchronicity of these events. For example, an early tropical warming has been described in recent studies related to (1) an early deglaciation of the Antarctic Peninsula (Weber et al., 2014) along with (2) an early rise in global sea level (Lambeck et al., 2014), leading to (3) early responses of the north hemispheric mid-latitude terrestrial and oceanic records (Jackson et al., 2019). (4) Early warming in tropical SST is also described before any considerable ice volume change took place in high latitudes (Romero et al., 2015). Therefore, we suggest that environmental responses to HS1 in low- and mid-latitudes could have preceded the climatic response of high-latitude regions.

With respect to the causes producing these "different environmental responses" triggering the early environmental response in our region, we could suggest different hypotheses. In a new version of the manuscript, we have included that the early European ice-sheet retreat and alpine glaciers melting from the Alps and Apennines at ca. 20,000 years BP suggest an early meltwater influx into the Western Mediterranean through the rivers from the northern Mediterranean borderlands (Bonneau et al., 2014). This conditions affected the Mediterranean Sea overturning circulation (Fink et al., 2015) and could have produced the early beginning of cold and arid conditions in this region, as it is the main factor controlling the climate in this area (Rohling et al., 2015). On the other hand, the enhanced MOW towards the Atlantic occurred ~15,500 years BP (Rogerson et al., 2010), several hundred years before the usually assumed end of HS1 (or the onset of the B-A) at ~14,700 yr BP. According to this, it could be hypothesized that the early enhanced MOW and the own Mediterranean Sea thermohaline reactivation could have previously affected climate at the lower latitude Mediterranean area, and could have caused an earlier environmental response of the warm and humid B-A interstadial in this region.

Nevertheless, the main goal of this manuscript is to show, describe and discuss the internal climatic variability of HS1 with its subdivision in 3 main phases and 7 sub-phases in the Padul-15-05 record, and not so much on the age discrepancies of HS1 with other paleoclimatic records.

Also, while the authors document the three-phased nature of HS1 they discuss neither the causes nor regional implications of the described features. Why does the division of HS1 matter at all?

As detailed in our Supplemental material, it has been known for some time that the HS1 was complex, perhaps with 3 parts. Here we show that it may have been even more complex than previously thought, and can be subdivided into 7 sub-phases based on the paleoenvironmental signal of the Padul record. In addition, we compared our record with other paleoclimate records from the studied area, showing a similar climatic pattern during HS1 (see Figure 3) and pointing to a regional feature. The possible causes of this variability are discussed in lines 202-219, mostly related with millennial- and centennial-scale climate variability forced by solar output variations and the influence of the North Atlantic polar front displacements in the land-ocean temperature contrast and in moisture advection between both environments. Thus, it might be important to look at other HSs in greater detail to see if those events are more complex as well.

In the new version of the manuscript we have also included that these relatively short-scale periods (such as Heinrich Stadials or Dansgaard/Oeschger events) and their smaller-scale internal oscillations observed with high-resolution data are important in order to understand how the environment reacts to fast climate changes and the causes affecting them (such as fast changes in the Mediterranean thermohaline circulation).

Further, the authors call for solar activity as a driver of changes in the Padul proxies. Can they elaborate on the exact mechanism? How solar activity translates to floral assemblage changes?

Solar activity has been proven to significantly affect past climate variability, which in turn affected vegetation dynamics. Solar activity changes could affect temperature, air and water masses and thus precipitation on land affecting plants (Tinner and Kaltenrieder, 2005; Lozano-García et al., 2007; Ramos-Román et al., 2016). In addition, Kofler et al. (2005) also suggested that solar output caused decreases in pollen production and treeline shifts in the Alps, which could also be affecting vegetation in the Sierra Nevada and, therefore, in the Padul area.

Last but not least, the spectral analyses results seem a bit at odds with common sense. 800 yr cyclicity during HS1 event of less than 3 ka duration is already suspicious. Periodicity of 2000 yr within HS1, BA and YD (line 232-233) is simply absurd! The YD itself only lasted for ca. 1000 yr.

Sorry for the misunderstanding. We did not want to say that the ~2000-yr cycle occurred during the HS1 itself (or YD itself) - we know that this would make no sense. We meant that this cycle can be identified from 20 to 11 cal ky BP, which includes the HS1, B-A and YD (Figs. 4a and 4c and Figs 5a-c). We have clarified this in the new version of the manuscript:

Lines 172-173: "The spectral analysis on xerophytes and *Ip* presented ~2000, 800 and 500-yr cycles from 20 to 11 cal ky BP (Figs. 4a and 4c)". We have also removed the 200-yr cycle (also from the Figure 4) as it is not representative due to the sampling resolution.

Lines 232-233: "The main periodicities obtained for climatic oscillations of ~2000 and 800 yrs between 20 and 11 cal kyr BP seem to be related to solar forcing."

We have also changed lines 183-185, clarifying that the maximum values of xerophytes during the 2000-yr cycle at ~18 and ~16 cal kyr BP would mark the HS1c and HS1a, whereas the minimum value at ~17 cal kyr BP would mark the HS1b (Figs. 5a and 5b).

Lines 191-192 have also been modified. We have clarified that the maximum values of the 800-yr cycle based on xerophytes at 18.2, 17.3, 16.5 and 15.8 cal kyr BP would mark HS1a.1, HS1a.3, HS1c.1 and HS1c.3, respectively. On the contrary, the minimum values of the 800-yr cycle at 17.7, 17 and 16.1 cal kyr BP suggest the HS1a.2, HS1b and HS1c.2, respectively (Figs. 5a and 5c). Coincident minima obtained in the 2000 and 800-yr cycles at ~17 cal kyr BP define the HS1b interval, a period of moderate humid conditions within HS1.

This 2000-yr periodicity is pervasive for the last 20 kyr BP, at least in Southern Iberia, associated to solar activity and/or monsoon activity (Rodrigo-Gámiz et al., 2014). Subsequent studies demonstrate that this "monsoon-like" signal is linked to Nile input (Bahr et al., 2015), transmitted by Mediterranean thermohaline circulation to western Europe (Kaboth-Bahr et al., 2018) and associated with winter precipitation for the last 1.3 Myr in Southern European lacustrine records (Lake Ohrid) (Wagner et al., 2019). The absence of this pervasive 2000-yr signal at the Padul-15-05 record (very similar to Lake Ohrid record) would be surprising, since both areas are very sensitive to winter precipitation synchronized with African monsoon.

*We have removed the Figure 4b as it is only showing the spectral analysis for the HS1, it is confusing and it is not providing additional information with respect to the Figure 4a.*

Paleoclimate research is so much more than tuning wiggly curves and finding prescribed periodicities in proxy records! Valid, original observations call for careful and thorough and original interpretations rather than preaching to the choir using empty but catchy phrases and unsubstantiated claims.

*We totally agree with this comment and that is exactly what we are trying to do in this study, avoiding tuning to other well-known records (such as those from Greenland), trying not to find prescribed periodicities as result of the tuning effect and avoiding circular reasoning.*

I wish the authors will address mentioned points in the revised version of their manuscript.

Best regards,

Ola Kwiecien

*We hope that our new manuscript version will be more convincing for Dr. Kwiecien. In any case we are thankful for the comments.*

**REFERENCES**

Bahr, A., Kaboth, S., Jiménez-Espejo, F. J., Sierro, F. J., Voelker, A. H. L., Lourens, L., Röhl, U., Reichart, G. J., Escutia, C., Hernández-Molina, F. J., Pross, J., and Friedrich, O.: Persistent monsoonal forcing of Mediterranean Outflow Water dynamics during the late Pleistocene, Geology, 43, 951-954, https://doi.org/10.1130/g37013.1, 2015.

Bonneau, L., Jorry, S. J., Toucanne, S., Silva Jacinto, R., and Emmanuel, L.: Millennial-scale response of a western Mediterranean river to late Quaternary climate changes: a view from the deep sea, The Journal of Geology, 122, 687-703, https://doi.org/10.1086/677844, 2014.

Camuera, J., Jiménez-Moreno, G., Ramos-Román, M. J., García-Alix, A., Toney, J. L., Anderson, R. S., Jiménez-Espejo, F., Kaufman, D., Bright, J., and Webster, C.: Orbital-scale environmental and climatic changes recorded in a new ~ 200,000-year-long multiproxy sedimentary record from Padul, southern Iberian Peninsula, Quaternary Science Reviews, 198, 91-114, https://doi.org/10.1016/j.quascirev.2018.08.014, 2018.

Fink, H. G., Wienberg, C., De Pol-Holz, R., and Hebbeln, D.: Spatio-temporal distribution patterns of Mediterranean cold-water corals (Lophelia pertusa and Madrepora oculata) during the past 14,000 years, Deep Sea Research Part I: Oceanographic Research Papers, 103, 37-48, https://doi.org/10.1016/j.dsr.2015.05.006, 2015.

Jackson, M. S., Kelly, M. A., Russell, J. M., Doughty, A. M., Howley, J. A., Chipman, J. W., Cavagnaro, D., Nakileza, B., and Zimmerman, S. R. H.: High-latitude warming initiated the onset of the last deglaciation in the tropics, Science Advances, 5, https://doi.org/10.1126/sciadv.aaw2610, 2019.

Kaboth-Bahr, S., Bahr, A., Zeeden, C., Toucanne, S., Eynaud, F., Jiménez-Espejo, F., Röhl, U., Friedrich, O., Pross, J., Löwemark, L., and Lourens, L. J.: Monsoonal Forcing of European Ice-Sheet Dynamics During the Late Quaternary, Geophysical Research Letters, https://doi.org/10.1029/2018gl078751, 2018.

Kofler, W., Krapf, V., Oberhuber, W., and Bortenschlager, S.: Vegetation responses to the 8200 cal. BP cold event and to long-term climatic changes in the Eastern Alps: possible influence of solar activity and North Atlantic freshwater pulses, The Holocene, 15, 779-788, https://doi.org/10.1191/0959683605hl852ft, 2005.

Lambeck, K., Rouby, H., Purcell, A., Sun, Y., and Sambridge, M.: Sea level and global ice volumes from the Last Glacial Maximum to the Holocene, Proceedings of the National Academy of Sciences, 111, 15296, https://doi.org/10.1073/pnas.1411762111, 2014.

Lozano-García, M. d. S., Caballero, M., Ortega, B., Rodríguez, A., and Sosa, S.: Tracing the effects of the Little Ice Age in the tropical lowlands of eastern Mesoamerica, Proceedings of the National Academy of Sciences, 104, 16200, https://doi.org/10.1073/pnas.0707896104, 2007.

Ramos-Román, M. J., Jiménez-Moreno, G., Anderson, R. S., García-Alix, A., Toney, J. L., Jiménez-Espejo, F. J., and Carrión, J. S.: Centennial-scale vegetation and North Atlantic Oscillation changes during the Late Holocene in the southern Iberia, Quaternary Science Reviews, 143, 84-95, https://doi.org/10.1016/j.quascirev.2016.05.007, 2016.

Rodrigo-Gámiz, M., Martínez-Ruiz, F., Rodríguez-Tovar, F. J., Jiménez-Espejo, F. J., and Pardo-Igúzquiza, E.: Millennial- to centennial-scale climate periodicities and forcing mechanisms in the westernmost Mediterranean for the past 20,000 yr, Quaternary Research, 81, 78-93, https://doi.org/10.1016/j.yqres.2013.10.009, 2014.

Rogerson, M., Colmenero-Hidalgo, E., Levine, R., Rohling, E., Voelker, A., Bigg, G. R., Schönfeld, J., Cacho, I., Sierro, F., and Löwemark, L.: Enhanced Mediterranean-Atlantic exchange during Atlantic freshening phases, Geochemistry, Geophysics, Geosystems, 11, https://doi.org/10.1029/2009GC002931, 2010.

Rohling, E., Marino, G., and Grant, K.: Mediterranean climate and oceanography, and the periodic development of anoxic events (sapropels), Earth-Science Reviews, 143, 62-97, https://doi.org/10.1016/j.earscirev.2015.01.008, 2015.

Romero, O. E., Kim, J. H., Bárcena, M. A., Hall, I. R., Zahn, R., and Schneider, R.: High-latitude forcing of diatom productivity in the southern Agulhas Plateau during the past 350 kyr, Paleoceanography, 30, 118-132, https://doi.org/10.1002/2014PA002636, 2015.

Tinner, W., and Kaltenrieder, P.: Rapid responses of high-mountain vegetation to early Holocene environmental changes in the Swiss Alps, Journal of Ecology, 93, 936-947, https://doi.org/10.1111/j.1365-2745.2005.01023.x, 2005.

Wagner, B., Vogel, H., Francke, A., Friedrich, T., Donders, T., Lacey, J. H., Leng, M. J., Regattieri, E., Sadori, L., Wilke, T., Zanchetta, G., Albrecht, C., Bertini, A., Combourieu-Nebout, N., Cvetkoska, A., Giaccio, B., Grazhdani, A., Hauffe, T., Holtvoeth, J., Joannin, S., Jovanovska, E., Just, J., Kouli, K., Kousis, I., Koutsodendris, A., Krastel, S., Lagos, M., Leicher, N., Levkov, Z., Lindhorst, K., Masi, A., Melles, M., Mercuri, A. M., Nomade, S., Nowaczyk, N., Panagiotopoulos, K., Peyron, O., Reed, J. M., Sagnotti, L., Sinopoli, G., Stelbrink, B., Sulpizio, R., Timmermann, A., Tofilovska, S., Torri, P., Wagner-Cremer, F., Wonik, T., and Zhang, X.: Mediterranean winter rainfall in phase with African monsoons during the past 1.36 million years, Nature, 573, 256-260, https://doi.org/10.1038/s41586-019-1529-0, 2019.

Weber, M. E., Clark, P. U., Kuhn, G., Timmermann, A., Sprenk, D., Gladstone, R., Zhang, X., Lohmann, G., Menviel, L., Chikamoto, M. O., Friedrich, T., and Ohlwein, C.: Millennial-scale variability in Antarctic ice-sheet discharge during the last deglaciation, Nature, 510, 134-138, https://doi.org/10.1038/nature13397, 2014.

---

## Referee Comment (RC3) · Anonymous Referee #3 · 6 Feb 2020

In this manuscript, Dr Jon Camuera and co-authors propose to divide the Heinrich Stadial 1 (HS1), one of the coldest and driest phases of the last glacial, on the basis of the changes in moisture availability and temperature inferred from the high-resolution pollen sequence of the Padul wetland (southern Spain). The authors also use sedimentological data to support their argumentation. The chronological framework is based on a number of radiocarbon dates from bulk sediment and classical age-depth modelling. The chronology of their record led the authors to suggest the HS1 in the western Mediterranean region to have an offset of ca. 1 ka with respect to the Greenland ice

cores and most (if not all) of the European well-dated sedimentary records currently available. The research questions addressed are timely and of great relevance for the palaeoclimatic community, the methods used are mostly appropriate, and I acknowledge the effort the authors have made to discuss their results in the context of previously published regional palaeoclimatic evidence. However, I have noticed several important issues that the authors should address to make their arguments more convincing. At the moment, after reading carefully the manuscript, I am afraid that the main conclusions of the research are not fully supported by the data presented. In the following paragraphs, I will elaborate on why I got this impression.

GENERAL COMMENTS

1. A high number of radiocarbon dates does not necessarily mean that the chronology is robust. In this case, as shown in Figure 2 and Table S2, the number of radiocarbon dates available is truly impressive but all the accepted ones are on organic bulk sediment and bulk carbonate. The risk of hard-water effect is notable when dating bulk sediment from lakes whose catchments are mostly on calcareous bedrock. This holds particularly true for sections of the sedimentary sequence with higher shares of inorganic matter. The bedrock in the Padul-Niguelas basin is mostly calcareous (see Ramos-Roman et al., 2018), which is relevant also for this work and should be indicated in the main text or at least in the SI. This would contribute to explain the offset of ca. 1000 years observed in the Padul record at the beginning and the end of the HS1 compared with most of the well-dated records from Greenland and elsewhere in Europe. To have a sound support for their conclusions in this regard, the authors should provide a chronology based on short-lived terrestrial plant macrofossils or try to find out what the reservoir age for this period is. In any case, I consider that it is important that the authors discuss the limitations of their chronology.

2. Although I also see some patterns in the pollen data and the synthetic pollen-based indices suggestive of changes in the local climate, the record is quite "noisy". I was wondering whether the authors have tried to check and validate their visual delimitation of the phases and sub-phases within the HS1 using numerical tools such as zonation. Divisive (e.g. optimal splitting) or agglomerative (e.g. CONISS) methods to delimit groups of samples with similar pollen assemblages in an objective manner (pollen zones) would be quite appropriate in this case because otherwise the position of the boundaries for the different periods seems to be somehow arbitrary.

3. I acknowledge the interest of using indices to summarize pollen data to facilitate their interpretation, but it is necessary to see the raw data (in the SI if not in the main text) because the abundances of specific taxa may be relevant for the palaeoclimatic interpretation of the results. I would kindly ask the authors to supply a pollen diagram for the relevant period under investigation so the readers can fully assess the significance of the dataset.

4. According to the authors, the HS1 lasted 2.8 kyr in Padul, which is also in agreement with other continental, marine and ice records. Then, do periodicities of 2000 or even 800 years make sense at all? To me, on due respect, perhaps not too much...

SPECIFIC COMMENTS

Several words are repeated in the title and the keywords, i.e. Climate, Heinrich Stadial 1 and western Mediterranean. Perhaps the authors might consider adding some different keywords to make their paper easy to be found in scientific databases, e.g. southern Iberia, palaeoclimate or pollen analysis.

L18 Replace "generating" with "characterized by"?

L39-40 The authors might consider adding here a very relevant reference about last glacial rapid climatic variability in the North Atlantic context: Sanchez Goni et al. (2008) Quat. Sci. Rev. 27, 1136-1151.

L46 Replace "focus" with "have focused"?

L47 Replace "short-scale" with "short-term"? I think it would be more adequate...

L49-53 At least part of the content of the SI should be moved here so the reader gets a better idea of the background that justifies this research.

L54 "Sedimentological" instead of "sedimentation"?

L58 There is a mismatch between the time resolution indicated here, i.e. "61-yr", and the values shown in the SI, i.e. "77-yr" and "131-yr". Please, check and be consistent.

L81-83 Low percentages of xerophytic pollen during the LGM are quite unexpected. I have been checking the pollen diagram in Camuera et al. (2019) Quat. Sci. Rev. and pines were particularly abundant by that time around Padul. I was wondering whether the authors could provide a plausible explanation for these values because I consider this point would be worth to be discussed.

As I already said in my general comments, I see the patterns that the authors point out but it is also true that there is a significant overlap in the values, especially around the boundaries. The authors could seek statistical support for their inferences using a zonation procedure. HS1a.1-HS1a.3 and HS1c.1-HS1c.3 are hard to accept. Perhaps increasing the number of samples would bring support to this proposal, but with the current record it is not possible to assess whether peaks in a single sample are palaeoclimatically meaningful or just outliers.

L129-153 The authors should discuss here the potential hard-water effect in their chronology.

L146-148 References supporting this statement are needed.

L150 "Conditions" could be deleted.

L159 How the authors would explain the ca. 1000-yr offset between the onset of the Lateglacial Interstadial at Padul and most well-dated European records, including those from southern Europe, e.g. Monticchio, Trifoglietti?

SUPPLEMENTARY INFORMATION

L21 Please delete "one of", because it is certainly the most recent HS.

Regional and local settings. It is highly relevant that the authors inform about the bedrock in the catchment.

Chronology. Please, indicate which dates are new and which were published in previous studies.

L93 For the future, it would probably be worth to discuss the presence of Carpinus.

Spectral analysis and filtering. At least part of this text should be moved to the main text.

---

## Author Comment (AC2) · 16 Mar 2020

General comments

The manuscript by Jon Camuera and colleagues describes paleoenvironmental changes in southern Iberia during the last deglaciation focusing on Heinrich Stadial 1 (HS1). The authors observe a novel subdivision of HS1 in the analyzed Padul record and in other records from the Western Mediterranean and Greenland. They come to the conclusion that solar forcing accounted for an detected ~2000 and ~800 yrs climate cyclicity.

The study presents novel ideas and addresses relevant questions within the scope of the journal Climate of the Past. It is well structured, easy to follow, and concisely written. Figures are of good quality.

However, I have two main concerns. Firstly, it is not always clear whether data is new, already published, or already published but analyzed/shown in a new way. This concerns mainly the own previous studies. Nevertheless, it is important to exactly indicate the sources to avoid (self-) plagiarism (see also specific comments).

Thank you for the comment. In this new version of the manuscript we have clarified that we increased the resolution of the palynological analysis with respect to the previous study by Camuera et al. (2019). In this new study we analyzed 24 additional pollen samples between 20 and 11 kyr BP, and therefore, increased the pollen resolution from 67 samples (Camuera et al., 2019) to 91 samples (now explained in the new version of the manuscript). This permitted us to focus on the environmental and climate variability during HS1.

Secondly, the age-depth model is not as robust as stated. That does probably not affect the observed climate pattern but it may affect the cyclostratigraphic analysis. The age-depth model is based on radiocarbon dates of organic bulk sediments that might need a reservoir correction. Particularly in a wetland setting, a reservoir age of aquatic and semi-aquatic plants must be considered. The uncertainties of the age-depth model need to be taken into account and should be critically discussed when correlating records and when analyzing cyclicities.

Thank you for the comments. Firstly, we would like to apologize for the mistake we made in Table S1 (Table S2 in the previous version). The dated material of the new six samples analyzed for this study are organic vegetal residues from peat and carbonate sediments, pretreated with HCl and HF before submission to the BETA analytic laboratory for dating, and not bulk sediment as we previously stated. The organic vegetal residue should be characterized by lower reservoir effect (if any) after removing inorganic carbonates and silicates. In addition, two samples that were considered too old and were not used in the age model in the previous version of the manuscript have also been included in the new age-depth model (Fig. 2 and Table S1 in the new version) to avoid subjectivity.

With respect to the comment about dating of aquatic and semi-aquatic plants, most of the vegetation surrounding the wetland is mainly characterized by *Phragmites australis*

and *Typha domingensis* that use the atmospheric carbon as source for the photosynthesis (Brix et al., 2001; Dong et al., 2012). This information about the present-day vegetation of the Padul wetland and surrounding areas have been included in the new version of the Supplementary Information (section *Regional and local settings*). The terrestrial origin of the plant material used for radiocarbon dating is confirmed by values in $\delta^{13}C$ and C/N that are in agreement with C3 vascular land plants (Meyers, 2003; Meyers and Lallier-vergés, 1999). The $\delta^{13}C$ and C/N values have been included in new Table S1. Therefore, according to the vegetation from the Padul wetland and the $\delta^{13}C$ and C/N values recorded from samples, we can suggest that the reservoir effect of the organic carbon from the dated organic plant remains should not be high. In any case, we are aware that our radiocarbon datings could be biased by a possible reservoir effect and discussion about this possible problem has been included in the new version of the manuscript.

In order to improve the age model, in this new version of the manuscript we built a new age model based on Bayesian modelling, which accounts better for age uncertainties. This new age model for the Padul-15-05 record has been run for the last 30,000 cal yrs BP using the already published radiocarbon dates (including three specific compound radiocarbon samples) (Camuera et al., 2018) along with the six new radiocarbon samples analyzed for this study. Therefore, the new Bayesian age-depth model for the last 30 kyr BP is based on 40 radiocarbon dates (Fig. 2 and Tables S1 and S2 in the new version of the manuscript). All this changes have been included in the new version of the manuscript.

Specific comments

23/62: Please relativize the terms "robust" and "accurate".

Thanks. We have changed and deleted these terms.

25: Please clarify which resolution is improved.

Done. We have clarified the total amount of pollen samples analyzed and how many of them were new samples analyzed for this study. We have also clarified the new pollen resolution taking into account the new pollen samples. This part has been modified from the Supplementary Information (section *Palynological analysis*) and moved to the main manuscript (section *Materials and methods*) as we think it is important information for the reader to understand the bases of this study.

34/35: Why does natural climatic variability underlie abrupt anthropogenic climate change? Please clarify or rephrase.

We have clarified the sentence.

55: Please mention the section "Regional and Local Settings" of the Supplementary Information here. In addition, please delete "new" to avoid misunderstanding.

Thank you for the suggestions. We have mentioned the "Regional and Local Settings" section and we have deleted the word "new".

62–64: Please add reference of the radiocarbon dates.

Thank you for the comment. However, we have added the sample references in Table S1 and thus we think that it is not necessary to add the references of the radiocarbon dates in the main text. In that new table S1 we also included the $\delta^{13}C$ and C/N values

from the six new radiocarbon samples analyzed for this study as well as the information from the previously analyzed and published samples from Camuera et al. (2018).

65/66: Please add reference of the pollen data, e.g. add "based on palynological data by…" after "Precipitation Index (Ip)". If I understood it right, the palynological data has already been published, but it is not clearly stated in the manuscript.

As we explained in your question "25: Please clarify which resolution is improved", we have clarified the total amount of pollen samples used and how many of them are new samples specifically done for this study. We have also clarified the new pollen resolution after analyzing the new samples.

81–83: How is the start (lower boundary) of HS1 defined in your record? Could it have also started at ca. 18.7 kyr when Si, Mediterranean forest, PCI and Ip start to decline?

Yes, HS1 could have started before according to the PCI and *Ip*. However, we think that even if the decline in the PCI and *Ip* started a few hundred years before, the really arid conditions characteristic of HS1 should have started at 18.4 kyr BP coinciding with the abrupt increase in xerophyte percentages. In addition, as both declining trends of PCI and *Ip* for the beginning of HS1 are relatively gradual/transitional (not as fast as during the end of HS1), we took a middle point from this declining trend coinciding with the abrupt increase in xerophytes.

97/98: Please add "(Fig. 3b, c)" after SST reconstructions and "Cacho et al., 1999; 2006" to the references.

Done.

102–104: Please rephrase the sentence because the SST records published by Cacho et al., 1999; 2006 originate from the Alboran Sea as well.

Yes, we agree. We have rewritten the sentence.

109–111: Please add PCI because it shows the same pattern.

Yes, we also agree. We have also included the PCI as it is also showing a general lowering trend related with a decreasing moisture pattern.

112: Please replace "(Fig. 3a, b)" by "(Fig. 3b)".

Thank you. Done.

136–157: The presented explanations and records are not strong enough to conclude an early HS1 and Bølling-Allerød in the Mediterranean.

We think the reviewer is right. We have proposed several possible causes that could have triggered the early environmental response in our region. However, we have also included uncertainties with the radiocarbon dating that could generate this early HS1 record in Padul (or even a combination of both factors: early environmental response and radiocarbon dating uncertainties).

159–169: Please add comparisons with other regional records.

We thank the reviewer for the suggestion. We have included references of other regional records presenting environmental reconstructions for the BA and YD periods. These records suggest climate conditions similar to our Padul record.

166–169: Xerophytes decrease at first. How can that be explained? How is the lower boundary of the YD defined in your record? Better mention the Ip value to suggest arid conditions. In general, it would be nice to see a detailed pollen diagram in the supplement to comprehend the stated climate variations.

Thank you for the comment. However, the lower boundary of the YD is not defined by xerophyte percentages but mainly by changes in the *Ip* values (as the reviewer suggested). Comparing xerophyte percentages (Fig. 3c of the new version of the manuscript) with respect to *Ip* values (Fig. 3e), we can observe that the lower boundary of the YD is well defined by a decrease in *Ip* at ca. 12.9 kyr BP, whereas xerophytes are not decreasing (note the inverse xerophyte values) until 12.6 kyr BP. We apologize for not including this in the previous version. We have changed this paragraph in order to better clarify the boundaries of the YD period.

In addition, we have also included a detailed pollen diagram with the most characteristic pollen taxa in the Supplementary Information (Fig. S1). The pollen diagram also shows the CONISS cluster analysis, which helped us to confirm the climate variability for this period deduced by the pollen data.

185–188: The D/O-record for 20-11 kyr is well defined and does not show a ~2 kyr cyclicity.

Thank you for the comment - we agree. Bond et al. (1999) suggested that the D/O cycles seem to be an amplification of the 1-2kyr cycle. Therefore, we have modified the phrase.

244: I appreciate that you provide the data in an online repository. However, I suggest adding the complete palynological dataset to the repository for replicability.

Yes, we agree. We have uploaded the complete palynological dataset to the PANGAEA data repository (https://doi.pangaea.de/10.1594/PANGAEA.904053, *dataset in review*).

Figure 2–5: Please indicate all sources of data.

Thanks for the comment. We have included the link of the data repository at the end of the Figure 3 caption (Fig. 2 in the previous version). Sources are included in Figure 4 (Fig. 3 in the previous version). With respect to the Figure 6 (Fig. 5 in the previous version), we have added the reference of the source of the [10]Be flux data (i.e., Adolphi et al., 2014), whereas the source of the pollen data from Padul record has already been included in the Figure 3 (as well as in the section *Data availability*).

Figure 2a: The uncertainty of the age-depth model is underestimated where no dates are available. Please use a model that accounts better for uncertainties. In addition, please add the dates that you rejected to Fig. 2a, e.g. in a different color.

We agree with the reviewer. In this new version of the manuscript we built a new Bayesian age-depth model to have a better control of age uncertainties and improve the Padul-15-05 age model for the last 30 kyr BP. Again, we would like to apologize for the misunderstanding concerning the dated materials for the six new radiocarbon samples, as they were not bulk sediments but organic plant residues. Therefore, the six new samples have also been included in the new Bayesian age-depth model. See the new Figure 2 and Tables S1 and S2 to see the new Bayesian age-depth model and included/rejected radiocarbon dates for the last 30 kyr BP.

Figure 4: I suggest to use always "xerophyte percentages" instead of "raw xerophyte data" and "raw xerophyte percentages" (also in Supplementary Information line 120). In

addition, please indicate the meaning of the green lines (confidence interval) in the figure caption. Which periodicity is shown by the first peak in Fig. 4d and why is it not mentioned?

Thank you for the suggestions. We have changed those expressions to "xerophyte percentages" and we have included the meaning of the green lines (confidence intervals) in the figure caption. The fist periodicity peak observed in Fig. 5b (Fig. 4d in the previous version) has to be an artefact as it shows a cycle with a periodicity between 1689 and 5068 years (frequencies between 0.0001973 and 0.0005919) for a time series of 7600 years ([10]Be flux data, from 18.6 to 11 kyr BP). This is explained at the end of the figure caption.

Supplementary Information (SI): The Supplementary Information is a rather extensive compilation of additional details. I appreciate the methodological details here. However, I suggest including the previous studies to the main text because they contain important data for comparison. For an even better comparison, I suggest adding this study to table S1.

Thanks for these constructive suggestions. The previous studies from Camuera et al. (2018, 2019) are included in the main text.

We totally agree in adding the interpretation of this study in new Table 1 (Table S1 in the previous version) along with the rest of the studies for a better comparison of environmental interpretations between studies/records. In addition, this Table 1 has been moved from the Supplementary Information to the main text.

Table S2: Please add source (reference or this study) to each date.

Thank you for the suggestion. We have added letters ([a] and [b]) for identifying samples analyzed in previous studies and those analyzed for this study.

SI 54: Please add one or two sentences about the recent vegetation.

Thanks for the comment. We agree. We have included the most characteristic vegetation from the Padul wetland and surrounding areas.

SI 91–93/100: Please indicate which taxa are mesothermic and which are steppic.

Ok, done.

SI 107–109: Is this new or already published data? Please clearly indicate.

Thank you for the suggestions. In the Supplementary Information we have clarified that the inorganic geochemical composition of the entire Padul-15-05 record (the last ca. 197 kyr BP) was published in Camuera et al. (2018) and was done to observe orbital-scale environmental changes for the last 2 glacial-interglacial cycles.

SI 120–125: Which parameters were used for the Ip analysis? Could you add Ip to the first sentence?

The parameters used for the xerophytes and *Ip* data were the same (value of 2 for segments parameter and value of 3 for the oversample parameter). Nevertheless, we have removed the spectral analysis of *Ip* data. *Ip* values from Figure 3e (Fig. 2e in the previous version) are represented in a logarithmic scale, resulting in an inaccurate spectral analysis. In addition, we have also removed the sentence (and the related figure) about the spectral analysis of xerophytes for the age period between 18.4 – 15.6 kyr BP

(only HS1), as it was not showing any additional information with respect to Figure 5a (Fig. 4a in the previous version).

SI 120–137: Why were exactly these datasets used? Why is there only one analysis for HS1?

We used the dataset from xerophytes as it is the most characteristic and abundant group of pollen taxa responding to environmental and climate oscillations in Padul during HS1. As the reviewer says, there is only one spectral analysis for HS1, using the xerophytes time series. However, we have removed this spectral analysis from xerophytes for the HS1, as it is not showing any additional information with respect to the spectral analysis for the age range between 20 and 11 kyr BP (new Figure 5a in this new version of the manuscript).

Technical corrections:

74: Please edit format of reference.

We don't know what is wrong with the format of this reference. However, we have revised all references and the format.

167: shown.

Thank you. Changed.

SI 125: Please add "(CI)" after "Confidence Interval".

Thanks. We have added (CI).

SI 129: analyses.

Thanks, but it is not necessary as we have changed the sentence.

**REFERENCES**

Adolphi, F., Muscheler, R., Svensson, A., Aldahan, A., Possnert, G., Beer, J., Sjolte, J., Björck, S., Matthes, K., and Thiéblemont, R.: Persistent link between solar activity and Greenland climate during the Last Glacial Maximum, Nature Geoscience, 7, 662, https://doi.org/10.1038/NGEO2225, 2014.

Bond, G. C., Showers, W., Elliot, M., Evans, M., Lotti, R., Hajdas, I., Bonani, G., and Johnson, S.: The North Atlantic's 1-2 kyr climate rhythm: relation to Heinrich events, Dansgaard/Oeschger cycles and the Little Ice Age, Mechanisms of global climate change at millennial time scales, 112, 35-58, https://doi.org/10.1029/GM112p0035, 1999.

Brix, H., Sorrell, B. K., and Lorenzen, B.: Are Phragmites-dominated wetlands a net source or net sink of greenhouse gases?, Aquatic Botany, 69, 313-324, https://doi.org/10.1016/S0304-3770(01)00145-0, 2001.

Camuera, J., Jiménez-Moreno, G., Ramos-Román, M. J., García-Alix, A., Toney, J. L., Anderson, R. S., Jiménez-Espejo, F., Kaufman, D., Bright, J., and Webster, C.: Orbital-scale environmental and climatic changes recorded in a new ~ 200,000-year-long multiproxy sedimentary record from Padul, southern Iberian Peninsula, Quaternary Science Reviews, 198, 91-114, https://doi.org/10.1016/j.quascirev.2018.08.014, 2018.

Camuera, J., Jiménez-Moreno, G., Ramos-Román, M. J., García-Alix, A., Toney, J. L., Anderson, R. S., Jiménez-Espejo, F., Bright, J., Webster, C., and Yanes, Y.: Vegetation and climate changes during the last two glacial-interglacial cycles in the western Mediterranean: A new long pollen record from Padul (southern Iberian Peninsula), Quaternary Science Reviews, 205, 86-105, https://doi.org/10.1016/j.quascirev.2018.12.013, 2019.

Dong, W., Shu, J., He, P., Ma, G., and Dong, M.: Study on the Carbon Storage and Fixation of Phramites autralis in Baiyangdian Demonstration Area, Procedia Environmental Sciences, 13, 324-330, https://doi.org/10.1016/j.proenv.2012.01.031, 2012.

Meyers, P. A., and Lallier-vergés, E.: Lacustrine Sedimentary Organic Matter Records of Late Quaternary Paleoclimates, Journal of Paleolimnology, 21, 345-372, https://doi.org/10.1023/A:1008073732192, 1999.

Meyers, P. A.: Applications of organic geochemistry to paleolimnological reconstructions: a summary of examples from the Laurentian Great Lakes, Organic Geochemistry, 34, 261-289, https://doi.org/10.1016/S0146-6380(02)00168-7, 2003.

---

## Author Comment (AC4) · 16 Mar 2020

In this manuscript, Dr Jon Camuera and co-authors propose to divide the Heinrich Stadial 1 (HS1), one of the coldest and driest phases of the last glacial, on the basis of the changes in moisture availability and temperature inferred from the high-resolution pollen sequence of the Padul wetland (southern Spain). The authors also use sedimentological data to support their argumentation. The chronological framework is based on a number of radiocarbon dates from bulk sediment and classical age-depth modelling. The chronology of their record led the authors to suggest the HS1 in the western Mediterranean region to have an offset of ca. 1 ka with respect to the Greenland ice cores and most (if not all) of the European well-dated sedimentary records currently available. The research questions addressed are timely and of great relevance for the palaeoclimatic community, the methods used are mostly appropriate, and I acknowledge the effort the authors have made to discuss their results in the context of previously published regional palaeoclimatic evidence. However, I have noticed several important issues that the authors should address to make their arguments more convincing. At the moment, after reading carefully the manuscript, I am afraid that the main conclusions of the research are not fully supported by the data presented. In the following paragraphs, I will elaborate on why I got this impression.

GENERAL COMMENTS

1. A high number of radiocarbon dates does not necessarily mean that the chronology is robust. In this case, as shown in Figure 2 and Table S2, the number of radiocarbon dates available is truly impressive but all the accepted ones are on organic bulk sediment and bulk carbonate. The risk of hard-water effect is notable when dating bulk sediment from lakes whose catchments are mostly on calcareous bedrock. This holds particularly true for sections of the sedimentary sequence with higher shares of inorganic matter. The bedrock in the Padul-Niguelas basin is mostly calcareous (see Ramos-Roman et al., 2018), which is relevant also for this work and should be indicated in the main text or at least in the SI. This would contribute to explain the offset of ca. 1000 years observed in the Padul record at the beginning and the end of the HS1 compared with most of the well-dated records from Greenland and elsewhere in Europe. To have a sound support for their conclusions in this regard, the authors should provide a chronology based on short-lived terrestrial plant macrofossils or try to find out what the reservoir age for this period is. In any case, I consider that it is important that the authors discuss the limitations of their chronology.

We thank the reviewer for these constructive comments and suggestions.

In this new version of the manuscript we have clarified the possible issues concerning the influence of the dissolved carbonate from bedrock in the radiocarbon dating that could have affected the chronology. In addition, in the previous version of the manuscript there was a mistake in the dated "Material" column from Table 1 (Table 2 in the previous version of the manuscript) that has been corrected. The "bulk carbonate" and "bulk peat" material from the six new radiocarbon samples were not bulk sediments. These six

samples were taken as bulk sediment material from the core, but there were pretreated with HCl and HF in the Stratigraphy and Paleontology department from the University of Granada before submission to the BETA analytic laboratory for $^{14}$C measuring. Therefore, the reservoir effect from these organic plant residues from both peat and carbonate lithologies does not take into account inorganic carbonate and the reservoir effect should be low(er). Nevertheless, we also understand that our age-depth model can present inaccuracies and the high number of dates does not assure us a perfect chronological control for the age delimitation of HS1. Therefore, in the new version of the manuscript, we have included that even if the age-depth model can present uncertainties, the early environmental record of HS1 in our latitude cannot be ruled out, and several factors and causes have been provided for supporting our hypothesis of an early record of HS1 in the study area.

In addition, a new Bayesian age-depth model has been carried out for the last 30 kyr BP, including the two previously rejected dates (from the six new samples analyzed for the period concerning the HS1) in order to reduce error uncertainties and improve the chronology. As explained above, these six new samples were pretreated and the dated material was organic plant residue, and therefore, there are no objective reasons to exclude any of these samples (new Figure 2, *Chronology* section from the Supplementary Information and Tables S1 and S2).

2. Although I also see some patterns in the pollen data and the synthetic pollen-based indices suggestive of changes in the local climate, the record is quite "noisy". I was wondering whether the authors have tried to check and validate their visual delimitation of the phases and sub-phases within the HS1 using numerical tools such as zonation. Divisive (e.g. optimal splitting) or agglomerative (e.g. CONISS) methods to delimit groups of samples with similar pollen assemblages in an objective manner (pollen zones) would be quite appropriate in this case because otherwise the position of the boundaries for the different periods seems to be somehow arbitrary.

Thanks for the suggestion. In this new version of the manuscript we added a plot with the detailed pollen diagram including the most characteristic pollen taxa from Padul (percentages of samples above 1%) for the age period between 20 and 11 kyr BP (new Figure S1). In addition, we ran a cluster analysis (CONISS) on the most important pollen taxa (i.e., *Quercus* total, *Olea*, *Pistacia*, Cupressaceae, *Artemisia* and Amaranthaceae) to group environmentally- and climatically-similar samples in the HS1 record using an objective statistical method (with the Tilia software). This is shown in new Figure S1. Note that the agglomerative CONISS methodology shows the same variability than visually, identifying 7 sub-phases within HS1. See Supplementary Information (*Palynological analysis* section) for more information.

3. I acknowledge the interest of using indices to summarize pollen data to facilitate their interpretation, but it is necessary to see the raw data (in the SI if not in the main text) because the abundances of specific taxa may be relevant for the palaeoclimatic interpretation of the results. I would kindly ask the authors to supply a pollen diagram for the relevant period under investigation so the readers can fully assess the significance of the dataset.

Thank you for the comments. We agree and in this new version of the manuscript we included the pollen diagram of the most characteristic pollen taxa (as explained above) (Figure S1). In addition, we also uploaded the complete pollen data to PANGAEA data repository (https://doi.pangaea.de/10.1594/PANGAEA.904053, *dataset in review*).

4. According to the authors, the HS1 lasted 2.8 kyr in Padul, which is also in agreement with other continental, marine and ice records. Then, do periodicities of 2000 or even 800 years make sense at all? To me, on due respect, perhaps not too much…

Sorry for the misunderstanding, but we did not want to say that the 2000- and 800-yr cycles only occurred during HS1. As the reviewer says, it would make no sense because HS1 only lasted 2800 years. We have clarified in the text that the spectral analysis was run on the xerophytes times series for the age period between 20 and 11 kyr BP. Therefore, the spectral analysis is showing the periodicities of these proxies for an age range lasting 9000 years. In addition, we have also removed the spectral analysis done on xerophytes only for HS1 (Fig. 4b in the previous version) and the spectral analysis of *Ip* data (Fig. 4c in the previous version) as they were not showing any additional information with respect to the Figure 4a (Fig. 5a in the new version) and can also produce misunderstandings.

SPECIFIC COMMENTS

Several words are repeated in the title and the keywords, i.e. Climate, Heinrich Stadial 1 and western Mediterranean. Perhaps the authors might consider adding some different keywords to make their paper easy to be found in scientific databases, e.g. southern Iberia, palaeoclimate or pollen analysis.

We totally agree. Thank you so much for the suggestions. We have modified some keywords.

L18 Replace "generating" with "characterized by"?

Changed.

L39-40 The authors might consider adding here a very relevant reference about last glacial rapid climatic variability in the North Atlantic context: Sanchez Goni et al. (2008) Quat. Sci. Rev. 27, 1136-1151.

We agree. We have included this reference.

L46 Replace "focus" with "have focused"?

Changed.

L47 Replace "short-scale" with "short-term"? I think it would be more adequate…

Ok. Replaced.

L49-53 At least part of the content of the SI should be moved here so the reader gets a better idea of the background that justifies this research.

Thank you for the suggestion. We think that including the environmental conditions of every phase of the HS1 (early, middle and late) from six different studies (including Padul) is too much information for the *Introduction* section. However, we have moved the Table (Table S1 in the previous version of the manuscript) to the main text (Table 1 in the new version), as we also think that this information is important to understand the background of the paper (as the reviewer suggests). We think that including this Table 1 in the main text could be enough, as it summarizes the text from the Supplementary Information.

L54 "Sedimentological" instead of "sedimentation"?

Yes, we agree. Changed.

L58 There is a mismatch between the time resolution indicated here, i.e. "61-yr", and the values shown in the SI, i.e. "77-yr" and "131-yr". Please, check and be consistent.

The 61-yr resolution was for the period between 18.4 and 15.6 kyr BP. In the SI the 77-yr resolution was for the period between 20 and 15.6 kyr BP and the 131-yr resolution for the period between 15.6 and 11 kyr BP. Accordingly, we have modified and simplified the main text in the *Introduction* and *Materials and methods* sections for a better understanding.

L81-83 Low percentages of xerophytic pollen during the LGM are quite unexpected. I have been checking the pollen diagram in Camuera et al. (2019) Quat. Sci. Rev. and pines were particularly abundant by that time around Padul. I was wondering whether the authors could provide a plausible explanation for these values because I consider this point would be worth to be discussed.

Thank you for the interest. Pons and Reille (1988) suggested that the decline in trees except *Pinus* could be triggered by very arid climate but under not extremely cold temperatures. In additions, the high *Pinus* occurrence could suggest less marked ocean advection and higher continental climate conditions. As explained in Camuera et al. (2019), *Pinus* abundance in this record seems to be related with treeline movements during transitions between warm/humid interglacial/interstadials and the coldest/most arid periods (such as the Heinrich Stadials) (see also Figure 9 from Camuera et al., 2019). Therefore, we can suggest that *Pinus* acted as transitional taxa between relatively warm and humid periods and the coldest and most arid climate phases.

As I already said in my general comments, I see the patterns that the authors point out but it is also true that there is a significant overlap in the values, especially around the boundaries. The authors could seek statistical support for their inferences using a zonation procedure. HS1a.1-HS1a.3 and HS1c.1-HS1c.3 are hard to accept. Perhaps increasing the number of samples would bring support to this proposal, but with the current record it is not possible to assess whether peaks in a single sample are palaeoclimatically meaningful or just outliers.

As explained above, we have carried out a cluster analysis using CONISS (Tilia software) on the pollen data from the studied time period (new Figure S1). This agglomerative statistical analysis supports the 7 sub-phases subdividing HS1 in the Padul record, validating our previous subdivision of the SH1 with a statistical method. We thank the reviewer for the suggestion, as it has been really useful to have a better and more robust statistical methodology for the validation of the subdivision of the HS1.

L129-153 The authors should discuss here the potential hard-water effect in their chronology.

Ok. In this new version we have added more discussion about the influence of the hard-water and reservoir effect on radiocarbon datings from lacustrine environments, and therefore, the possible limitation of our chronological control.

L146-148 References supporting this statement are needed.

References are in the following sentence, when showing the examples from the CAN speleothem (N Spain), lake Prespa (Macedonia, Albania, Greece) and MD99-2331 (NW Iberian margin) records.

L150 "Conditions" could be deleted.

Deleted.

L159 How the authors would explain the ca. 1000-yr offset between the onset of the Lateglacial Interstadial at Padul and most well-dated European records, including those from southern Europe, e.g. Monticchio, Trifoglietti?

We have included new paragraphs pointing into several causes that could have influenced the early record of HS1 in our mid-latitude region with respect to high-latitude areas. As explained above and clarified in the main text, the high-resolution dating methodology from Padul does not assure a perfect chronological control for the HS1. However, possible asynchronicities and early record of HS1 in this mid-latitude Mediterranean region as result of the provided/explained causes should not be ruled out.

SUPPLEMENTARY INFORMATION

L21 Please delete "one of", because it is certainly the most recent HS.

Ok. We agree.

Regional and local settings. It is highly relevant that the authors inform about the bedrock in the catchment.

Done. We have included information about the catchment.

Chronology. Please, indicate which dates are new and which were published in previous studies.

We have clarified with letters ([a] and [b]) in the Table S1 (Table S2 in the previous version of the manuscript) the new samples analyzed for this study and samples analyzed and published in the previous study (Camuera et al., 2018).

L93 For the future, it would probably be worth to discuss the presence of Carpinus.

We agree. In the future, we would like to discuss the presence of both *Carpinus* and *Abies* during the last two glacial-interglacial cycles in Padul, as they are really interesting taxa that disappeared in the Padul record during the last interglacial period and the last glaciation, respectively.

Spectral analysis and filtering. At least part of this text should be moved to the main text.

Thanks for the suggestion. We have moved an important part of the spectral analysis to the main text, including the age range of the xerophyte data, periodicities, confidence intervals (and frequencies) and the software used.

**REFERENCES**

Camuera, J., Jiménez-Moreno, G., Ramos-Román, M. J., García-Alix, A., Toney, J. L., Anderson, R. S., Jiménez-Espejo, F., Kaufman, D., Bright, J., and Webster, C.: Orbital-scale environmental and climatic changes recorded in a new ~ 200,000-year-long multiproxy sedimentary record from Padul, southern Iberian Peninsula, Quaternary Science Reviews, 198, 91-114, https://doi.org/10.1016/j.quascirev.2018.08.014, 2018.

Camuera, J., Jiménez-Moreno, G., Ramos-Román, M. J., García-Alix, A., Toney, J. L., Anderson, R. S., Jiménez-Espejo, F., Bright, J., Webster, C., and Yanes, Y.: Vegetation and climate changes during the last two glacial-interglacial cycles in the western Mediterranean: A new long pollen record from Padul (southern Iberian Peninsula), Quaternary Science Reviews, 205, 86-105, https://doi.org/10.1016/j.quascirev.2018.12.013, 2019.

Pons, A., and Reille, M.: The Holocene and Upper Pleistocene Pollen Record from Padul (Granada, Spain): A new study, Palaeogeography, Palaeoclimatology, Palaeoecology, 66, 243-263, https://doi.org/10.1016/0031-0182(88)90202-7, 1988.

---

## Author Comment (AC5) · 16 Mar 2020

This paper presents the details of the recently published new Padul pollen record for the Heinrich Stadial 1 and Lateglacial interval.

The pollen record reveals significant changes over the study interval, presented in the form of pollen-based indices with established use in the study area. The record is also at a high temporal resolution offering new insights into centennial-scale variability during the study interval.

There are fascinating visual parallels between the pollen indices for the Heinrich Stadial 1 interval and SST records from the nearby W Mediterranean (Alboran Sea) which generally support the interest and interpretation of rapid climate variability during this interval.

The main difficulty for the manuscript is the chronology of the record. In essence, the Heinrich Stadial appears "too old" in the Padul record, and this creates difficulties for the analysis and interpretation. The manuscript seems to have a "split personality" – attempting to interpret both (A) the difference in ages between the Padul record and other records as a real and meaningful phenomenon, e.g. with implications for reservoir ages, etc. and (B) propose synchronicity of events between Greenland and S Iberia, e.g. as shown by wiggle-matched records in some figures and direct labelling of pollen changes with Greenland event stratigraphical terminology. It should be noted that conceptually (A) and (B) are mutually exclusive and they sit together very uncomfortably in the manuscript.

Regarding (A), the authors suggest that changes in marine reservoir effects might explain the difference in apparent age of the Heinrich stadial between Padul and the Iberian margin records. However, the logic is reversed here and the apparently older age of the Padul record cannot be explained away by marine reservoir effects which would tend to give older ages in the marine realm, not the terrestrial. Furthermore, the study of coupled land-sea tracers in nearby Alboran records (Comboureiu Nebout et al., 2009; Fletcher et al., 2010) already reveals a synchronous (within age model uncertainty) coupling of climate changes over the W Mediterranean and the high-latitudes, with possible modest enhancements of up to ~200 years of the marine reservoir effect.

We thank the reviewer for the comments. We agree about the reservoir effect. The explanation about the reservoir effect in marine environments makes no sense and we have removed this sentence from the manuscript. However, we still think that even if the high-resolution chronological dating from Padul does not assure us a perfect age control, the possible asynchronicities between our mid-latitude region and high-latitude areas could have occurred and cannot be ruled out.

However, the reviewer suggest that the studies of coupled land-sea tracers in the nearby Alboran Sea from Combourieu Nebout et al. (2009) (ODP 976 record) and (Fletcher et al., 2010) (MD95-2043 record) reveal synchronous climate changes between the W Mediterranean and the high-latitudes and we partly disagree. If one looks in detail at the

high-resolution proxy data from these two above-mentioned records and the high-resolution data from the 293G record from Rodrigo-Gámiz et al. (2011, 2014) (also from the Alboran Sea), one can see that there is not an exact synchronicity between these W Mediterranean records and the high-latitude Greenland records (climatically equivalent GS-2.1a dated at 17,480 – 14,692 yr BP according to Rasmussen et al, 2014) (see also Figure below).

Fletcher et al. (2010) suggested the beginning of this cold and arid HS1 period at 17.5 kyr BP and the end at 14.9 kyr BP in the MD95-2043 record from Alboran Sea. However, if we focus on the alkenone SST reconstruction from the same sedimentary record, it seems that the most drastic temperature decrease related with the beginning of this cold period could have occurred hundred years before, at ~18.1 kyr BP (see Figure below).

The 293G record from Rodrigo-Gámiz et al. (2011, 2014) (Alboran Sea) also displays an early HS1 response in this region, at least for the beginning of HS1, with changing values in both SST and Zr/Al at 17.9 kyr (see Figure below). This suggest that, according to this record, the environmental response during the onset of the HS1 in Alboran Sea was ~400 years earlier than in the high-latitude Greenland records. The end of HS1 is not as clear as the beginning, as the LDI-SST data is showing a transitional increase from 15.8 to 14.7 kyr BP and the change in Zr/Al is not as sharp as during the onset. According to their study, Rodrigo-Gámiz et al. (2011) placed the end of HS1 and beginning of Bølling-Allerød at 14.7 kyr BP, which seems to be approximately concordant with the end of the increasing trend in SST at 14.8 kyr BP and the end of the decreasing trend in Zr/Al (note inverted values) from ~15 to 14.8 kyr BP.

With respect to the ODP-976 record (Alboran Sea), the alkenone SST data show HS1 between 17.1 and 14.7 kyr BP, presenting differences with respect to the MD95-2043, 293G and Greenland records, especially for the beginning of the HS1 (see Figure below).

According to this, we still suggest that the environmental response of HS1 in our region does not seems to be exactly synchronous with respect to high-latitude areas. Several studies suggested early environmental responses in low- and mid-latitude areas during HS1 and deglaciation (early alpine glaciers melting, early Mediterranean Sea overturning circulation… e.g., Bonneau et al., 2014; Fink et al., 2015) (see the new version of the manuscript), but age uncertainties related with radiocarbon dating in marine and continental records cannot be ruled out. Therefore, age offset between mid- and high-latitude records may have been result of 1) the different environmental responses depending on the latitude/location, 2) imprecision in the chronology related to radiocarbon dating, or even 3) the sum of both factors. Further studies should be focused on these age differences between records at different latitudes in order to understand possible asynchronicities related with different environmental responses due to possible specific regional environmental conditions. Nevertheless, the main goal of this study is not to observe asynchronicities between different regions and latitudes but to show the environmental and climate conditions occurring in the southern Iberian Peninsula during the deglaciation and, for the first time, to describe the internal climatic variability of HS1 in seven sub-phases.

[Figure]

**Figure.** Different proxy records from the western Mediterranean. From top to bottom: Padul (dark blue), MD95-2043 (light blue), 293G (dark blue) and ODP-976 (light blue). The d18O record from NGRIP is represented in black. Green dots (and vertical dashed grey lines) show the age boundaries of HS1 (climatically correspondent GS-2.1a in Greenland) suggested by the correspondent studies.

Overall, I suspect that there are uncertainties in the site-specific age model which are not dealt with fully in the manuscript. Essential information for the validity of this study about stratigraphy, age control data and rejected dates must be included and discussed in the main manuscript and not placed in the supplementary material. Inspecting the radiocarbon data, it is evident that there are difficulties with reservoir ages or old carbon sources leading the authors to reject several dates obtained on bulk carbonates and gastropods. However, I do not see that it can be excluded that old carbon effects are not impacting also on the included dates made on bulk sediment. The authors need to deal with this more directly in the presentation of the record and ultimately the interpretation of the data. If the uncertainties in the age model are too great to support (A) then this shortcoming should be accepted and the implications of (B) can still be tentatively explored.

Thank you. We really appreciate these comments.

Firstly, we would like to apologize for the mistake about the dated material. In the Table S1 (Table S2 in the previous version of the manuscript), we said that samples were "bulk carbonate" and "bulk peat" and this was wrong. The six new additional radiocarbon samples that we used to improve the age-depth model for HS1 were taken from bulk carbonate and bulk peat lithologies, but were pretreated with HCl and HF in the laboratory of the Stratigraphy and Paleontology department at the University of Granada. Therefore, after removing the inorganic carbonates from both peat and carbonate sediments, the final residue sent to the BETA analytic laboratory was the organic plant residue, and not the bulk sediment itself. The two previously considered "too old" radiocarbon samples that were not used in the previous version of the manuscript have been included in the new Bayesian age-depth model, as all the six new radiocarbon

samples analyzed should have similar reservoir ages and we do not have objective reasons to exclude them.

In the new Table S1 (Table S2 in the previous version) we have also included the $\delta^{13}C$ and C/N values from the used radiocarbon samples in order to understand the origin of the organic carbon source. The observed values (for both $\delta^{13}C$ and C/N) seem to be in agreement with vascular C3 land plants (Meyers, 2003; Meyers and Lallier-vergés, 1999), suggesting that the organic carbon source should be principally atmospheric and the reservoir effect related with the organic carbon from plants should not be excessively high. Nevertheless, as explained above, we are aware that the high-resolution radiocarbon dating from the Padul record does not assure a perfect chronological control, and this possible "old carbon" problem has also been taken into account in the new version of the manuscript.

Moreover, we built a new age-depth model based on Bayesian modeling for the last 30 kyr BP in order to have a better control of the age uncertainties, including all the new six AMS radiocarbon samples analyzed for this study and the radiocarbon samples already published in Camuera et al. (2018). The new Bayesian age-depth model used in this new version of the manuscript is based on 40 radiocarbon dates (including three specific compound radiocarbon samples). All this information and related changes have been included in the new version of the manuscript, and Figure and Tables have also been modified (new Figure 2, *Methods* from Supplementary Information and Tables S1 and S2).

Without a more open and direct appraisal of the age control issue, I do not think that the time series analysis can be sustained. Although there do appear to be interesting pseudo-cyclical patterns in the proxies for some time intervals, the authors must be cautious about over-interpreting weak spectral signals (e.g. at 80%, 90% confidence levels) and cautious about identifying spectral peaks at high frequencies occurring close to three times the sampling resolution which may be spurious.

Thank you so much for the suggestions. We totally agree with the reviewer about identifying spectral peaks at high frequency and close to 2-3 times the sampling resolution. Therefore, the spectral peaks at ~200 years have been removed, as they are not statistically significant and do not provide with additional information for the main goal of the manuscript. In addition, we have also removed the spectral analyses of xerophytes for the HS1 and *Ip* data, as they are not show any additional information with respect to the spectral analysis shown by xerophyte pollen data between 20 and 11 kyr BP. Please see these changes in Figure 5 (Fig. 4 in the previous version).

**REFERENCES**

Bonneau, L., Jorry, S. J., Toucanne, S., Silva Jacinto, R., and Emmanuel, L.: Millennial-scale response of a western Mediterranean river to late Quaternary climate changes: a view from the deep sea, The Journal of Geology, 122, 687-703, https://doi.org/10.1086/677844, 2014.

Camuera, J., Jiménez-Moreno, G., Ramos-Román, M. J., García-Alix, A., Toney, J. L., Anderson, R. S., Jiménez-Espejo, F., Kaufman, D., Bright, J., and Webster, C.: Orbital-scale environmental and climatic changes recorded in a new ~ 200,000-year-long multiproxy sedimentary record from Padul, southern Iberian Peninsula, Quaternary Science Reviews, 198, 91-114, https://doi.org/10.1016/j.quascirev.2018.08.014, 2018.

Combourieu Nebout, N., Peyron, O., Dormoy, I., Desprat, S., Beaudouin, C., Kotthoff, U., and Marret, F.: Rapid climatic variability in the west Mediterranean during the last 25 000 years from high resolution pollen data, Climate of the Past, 5, 503-521, https://doi.org/10.5194/cp-5-503-2009, 2009.

Fink, H. G., Wienberg, C., De Pol-Holz, R., and Hebbeln, D.: Spatio-temporal distribution patterns of Mediterranean cold-water corals (Lophelia pertusa and Madrepora oculata) during the past 14,000 years, Deep Sea Research Part I: Oceanographic Research Papers, 103, 37-48, https://doi.org/10.1016/j.dsr.2015.05.006, 2015.

Fletcher, W. J., Sánchez Goñi, M. F., Peyron, O., and Dormoy, I.: Abrupt climate changes of the last deglaciation detected in a Western Mediterranean forest record, Climate of the Past, 6, 245-264, https://doi.org/10.5194/cp-6-245-2010, 2010.

Meyers, P. A., and Lallier-vergés, E.: Lacustrine Sedimentary Organic Matter Records of Late Quaternary Paleoclimates, Journal of Paleolimnology, 21, 345-372, https://doi.org/10.1023/A:1008073732192, 1999.

Meyers, P. A.: Applications of organic geochemistry to paleolimnological reconstructions: a summary of examples from the Laurentian Great Lakes, Organic Geochemistry, 34, 261-289, https://doi.org/10.1016/S0146-6380(02)00168-7, 2003.

Rasmussen, S. O., Bigler, M., Blockley, S. P., Blunier, T., Buchardt, S. L., Clausen, H. B., Cvijanovic, I., Dahl-Jensen, D., Johnsen, S. J., Fischer, H., Gkinis, V., Guillevic, M., Hoek, W. Z., Lowe, J. J., Pedro, J. B., Popp, T., Seierstad, I. K., Steffensen, J. P., Svensson, A. M., Vallelonga, P., Vinther, B. M., Walker, M. J. C., Wheatley, J. J., and Winstrup, M.: A stratigraphic framework for abrupt climatic changes during the Last Glacial period based on three synchronized Greenland ice-core records: refining and extending the INTIMATE event stratigraphy, Quaternary Science Reviews, 106, 14-28, https://doi.org/10.1016/j.quascirev.2014.09.007, 2014.

Rodrigo-Gámiz, M., Martínez-Ruiz, F., Jiménez-Espejo, F. J., Gallego-Torres, D., Nieto-Moreno, V., Romero, O., and Ariztegui, D.: Impact of climate variability in the western Mediterranean during the last 20,000 years: oceanic and atmospheric responses, Quaternary Science Reviews, 30, 2018-2034, https://doi.org/10.1016/j.quascirev.2011.05.011, 2011.

Rodrigo-Gámiz, M., Martínez-Ruiz, F., Rampen, S. W., Schouten, S., and Sinninghe Damsté, J. S.: Sea surface temperature variations in the western Mediterranean Sea over the last 20 kyr: A dual-organic proxy (UK′37 and LDI) approach, Paleoceanography, 29, 87-98, https://doi.org/10.1002/2013pa002466, 2014.

Wolff, E. W., Chappellaz, J., Blunier, T., Rasmussen, S. O., and Svensson, A.: Millennial-scale variability during the last glacial: The ice core record, Quaternary Science Reviews, 29, 2828-2838, https://doi.org/10.1016/j.quascirev.2009.10.013, 2010.